# Myeloid apolipoprotein E controls dendritic cell antigen presentation and T cell activation

Fabrizia Bonacina [1], David Coe[2], Guosu Wang[2], Maria P. Longhi [2], Andrea Baragetti[1,3], Annalisa Moregola[1], Katia Garlaschelli[3], Patrizia Uboldi[1], Fabio Pellegatta[3], Liliana Grigore[3], Lorenzo Da Dalt[1], Andrea Annoni[4], Silvia Gregori [4], Qingzhong Xiao[2], Donatella Caruso [1], Nico Mitro [1], Alberico L. Catapano[1,5], Federica M. Marelli-Berg[2] & Giuseppe D. Norata [1,3]

Cholesterol homeostasis has a pivotal function in regulating immune cells. Here we show that apolipoprotein E (apoE) deficiency leads to the accumulation of cholesterol in the cell membrane of dendritic cells (DC), resulting in enhanced MHC-II-dependent antigen presentation and CD4$^+$ T-cell activation. Results from WT and apoE KO bone marrow chimera suggest that apoE from cells of hematopoietic origin has immunomodulatory functions, regardless of the onset of hypercholesterolemia. Humans expressing apoE4 isoform ($\varepsilon$4/3–$\varepsilon$4/4) have increased circulating levels of activated T cells compared to those expressing WT apoE3 ($\varepsilon$3/3) or apoE2 isoform ($\varepsilon$2/3–$\varepsilon$2/2). This increase is caused by enhanced antigen-presentation by apoE4-expressing DCs, and is reversed when these DCs are incubated with serum containing WT apoE3. In summary, our study identifies myeloid-produced apoE as a key physiological modulator of DC antigen presentation function, paving the way for further explorations of apoE as a tool to improve the management of immune diseases.

[1] Department of Pharmacological and Biomolecular Sciences (DisFeB), Università Degli Studi di Milano, Milan 20133, Italy. [2] William Harvey Research Institute, Queen Mary University of London, London EC1M 6BQ, UK. [3] SISA Centre, Bassini Hospital, Cinisello Balsamo 20092, Italy. [4] San Raffaele Telethon Institute for Gene Therapy (SR-Tiget), IRCCS San Raffaele Scientific Institute, Milan 20132, Italy. [5] IRCSS Multimedica, Milan 20138, Italy. These authors contributed equally: Fabrizia Bonacina, David Coe. Correspondence and requests for materials should be addressed to G.D.N. (email: danilo.norata@unimi.it)

Cellular and systemic metabolism regulates the physiological and pathological functions of lymphocytes and other subsets of leukocytes[1,2]. Several lines of evidence indicate a key role of cholesterol in the regulation of immune responses which are not only associated with an increased demand for membrane synthesis during cell expansion, but also relate to the ability of cholesterol to engage type I interferon signaling[3]. This effect, in turn, supports cytotoxic T-cell effector function[4] and promotes lymphocyte proliferation induced by antigen-presenting dendritic cells[5]. Cholesterol is also a key constituent of lipid rafts, specialized microdomains of the cell membrane where, among others, toll-like receptors (TLRs), major histocompatibility complex (MHC) molecules, T-cell receptor (TCR) and B-cell receptor (BCR) are enriched[6–8]. Changes in cholesterol content modify raft-dependent signaling due to protein delocalization and impact immune cell functions[9–12]. Low cellular cholesterol content activates sterol receptor element binding protein (SREBP), a transcription factor which controls the expression of genes involved in cholesterol biosynthesis and uptake[13,14]. In contrast, the last step precursors of cholesterol biosynthesis, such as desmosterol, or products of cholesterol oxidation such as oxysterols, inhibit SREBP activity and activate liver X receptors (LXR) to favor cholesterol elimination from cells. Of note, LXR signaling has been proposed to couple sterol metabolism to T-cell proliferation in the adaptive immune responses. Indeed, LXR dependent ATP-binding cassette subfamily G member 1 (ABCG1), promoting cholesterol efflux from cells to lipoproteins, limits T-cell proliferation[15]. Vice versa intracellular cholesterol accumulation, as a consequence of ABCA1 and ABCG1 deficiency results in leukocytosis and the expansion of progenitor cell populations in mice[16].

Classically, hypercholesterolemia has been indicated as the driver of such metabolic alterations occurring in immune cells. ApoE KO or LDLR KO mice fed an atherogenic diet develop pronounced hypercholesterolemia and display an immune-activated phenotype characterized by increased T-effector memory cells, which mimics the profile observed in hypercolesterolemic patients[17]. In the same experimental settings, the overexpression of apolipoprotein A-I (apoA-I), which increases the ability to transport cholesterol back to the liver, results in a reduced cellular cholesterol accumulation and immune cell activation in lymph nodes[18,19]. These data point to a critical role for apolipoproteins, including apoA-I and apoE, in controlling cholesterol immunometabolism at both a systemic and cellular level.

ApoA-I is mainly synthesized by the liver and the intestine, while apoE derives mainly from the liver, but is also produced by myeloid cells[20]. While hepatic derived apoE is associated to very low density lipoprotein (VLDL) and contributes to their catabolism, leading to atherosclerosis in apoE KO mice, myeloid-derived apoE is present on nascent HDL. Of note, apoE is also found on the surface of hematopoietic stem and multipotent progenitor cells (HSPCs) in a proteoglycan-bound pool, where it appears to control cell proliferation in an ABCA1- and ABCG1-dependent fashion, causing monocytosis in apoE KO mice[21]. Moreover, apoE was reported to modulate neutrophil and macrophage activation[22,23], worsening the prognosis of *Listeria monocytogenes* or *Klebsiella pneumoniae* infections[24], to facilitate lipid antigen presentation by CD1 molecules to natural killer T cells (NKT)[25] and to increase susceptibility to experimental autoimmune encephalomyelitis[26]. ApoE KO mice showed increased T-cell infiltration of the vascular wall[27] and increased circulating levels of T-effector memory cells[17], pointing to an increased activation of the adaptive immune response as a result of apoE deficiency. However, the molecular mechanisms leading to the immunomodulatory role of apoE on adaptive immunity has not been fully elucidated.

Here we investigate the immunomodulatory role of apoE with a major focus on the regulation of cholesterol homeostasis in cells involved in the adaptive immune response. Our results from experimental models and humans reveal a critical function of myeloid-derived apoE in controlling DC antigen presentation and T-cell priming. They further indicate that this apoE function is mediated through the autocrine/paracrine modulation of cholesterol metabolism in DCs, and is independent of systemic hypercholesterolemia.

## Results

**ApoE deficiency boosts CD4$^\pm$ T-cell-mediated immune response**. An initial assessment of the immune phenotype of apoE KO mice revealed that these mice display splenomegaly (Fig. 1a, b) with an accumulation of CD4$^+$CD44$^{hi}$CD62L$^{lo}$ T cells (effector memory, $T_{EM}$) but not of CD8$^+$CD44$^{hi}$CD62L$^{lo}$ in the spleen (Fig. 1c, d and Supplementary Figure 1A), and a concomitant decrease of CD4$^+$CD44$^{lo}$CD62L$^{hi}$ T cells ($T_{naive}$). In addition, the proportion of activated CD44$^{hi}$CXCR3$^{hi}$ CD4$^+$, but not of CD8$^+$ T cells increased in the spleen of apoE KO mice compared to their WT counterparts (Fig. 1e and Supplementary Figure 1B). A similar profile was observed in the blood and in peripheral lymph nodes (Supplementary Figure 1C-H). No significant differences in CD4$^+$CD44$^{hi}$CD62L$^{hi}$ (central memory, $T_{CM}$) (Fig. 1c, d) and in CD4$^+$CD25$^{hi}$FOXP3$^+$ T-regulatory cells (Treg) (Supplementary Figure 2A) were observed in the circulation and in secondary lymphoid organs between WT and apoE KO mice. To investigate the physiological relevance of this phenotype, we analyzed the survival of B6K$^d$-derived skin grafts transplanted onto WT and apoE KO mice (Fig. 1f), an alloresponse mediated by both CD8$^+$ and CD4$^+$ T cells[28]. As the impact of apoE deficiency was mainly observed on CD4$^+$ T cells, all recipients received a depleting anti-CD8 antibody prior to transplantation[29] to transiently remove the CD8$^+$-mediated response and allow activation of CD4$^+$ T cells with indirect allospecificity. ApoE deficiency led to significantly faster allograft rejection (Fig. 1g), increased CD4$^+$ T cells in the draining lymph nodes, expansion of CD4$^+$CD44$^{hi}$ and CD4$^+$CD44$^{hi}$CXCR3$^{hi}$ cells (Fig. 1h–j) and polarization toward CD4$^+$ $T_{EM}$ cells (Supplementary Figure 3A). In line with these observations, T cells isolated from lymph nodes of apoE KO mice had an increased migratory capacity towards CXCL10 (Fig. 1k), which stimulates the migration of CXCR3$^+$ effector T cells[30] when compared to wild-type T cells. In contrast, there was no difference in migration towards CCL19 and CCL21, which stimulates the migration of CCR7$^+$ naive and $T_{CM}$ cells[31] (Fig. 1l) and in basal migration (Supplementary Figure 3B). These data indicate that apoE deficiency results in enhanced CD4$^+$ T-cell proliferation and biased differentiation toward an effector memory phenotype.

**Hematopoietic-derived apoE regulates CD4$^\pm$ T-cells activation**. ApoE is a component of certain plasma lipoproteins and mediates lipoprotein catabolism in the liver, mainly by binding the VLDL-receptor, the LDL-receptor and the apoE2 receptor on the hepatocyte surface[32,33]. Therefore, apoE deficiency is associated with a decreased catabolism of atherogenic lipoproteins thus favoring hypercholesterolemia and atherosclerosis development[34,35]. We and others have shown that hypercholesterolemia induces systemic inflammation, which is reflected in increased circulating T-effector cells in mice and in humans[17,36,37]. Therefore, the phenotype observed in apoE KO mice might results from systemic inflammation driven by dyslipidemia. To address this possibility, we transplanted B6K$^d$-derived skin allografts onto WT or apoE KO mice that had been lethally irradiated and reconstituted with bone marrow (BM)

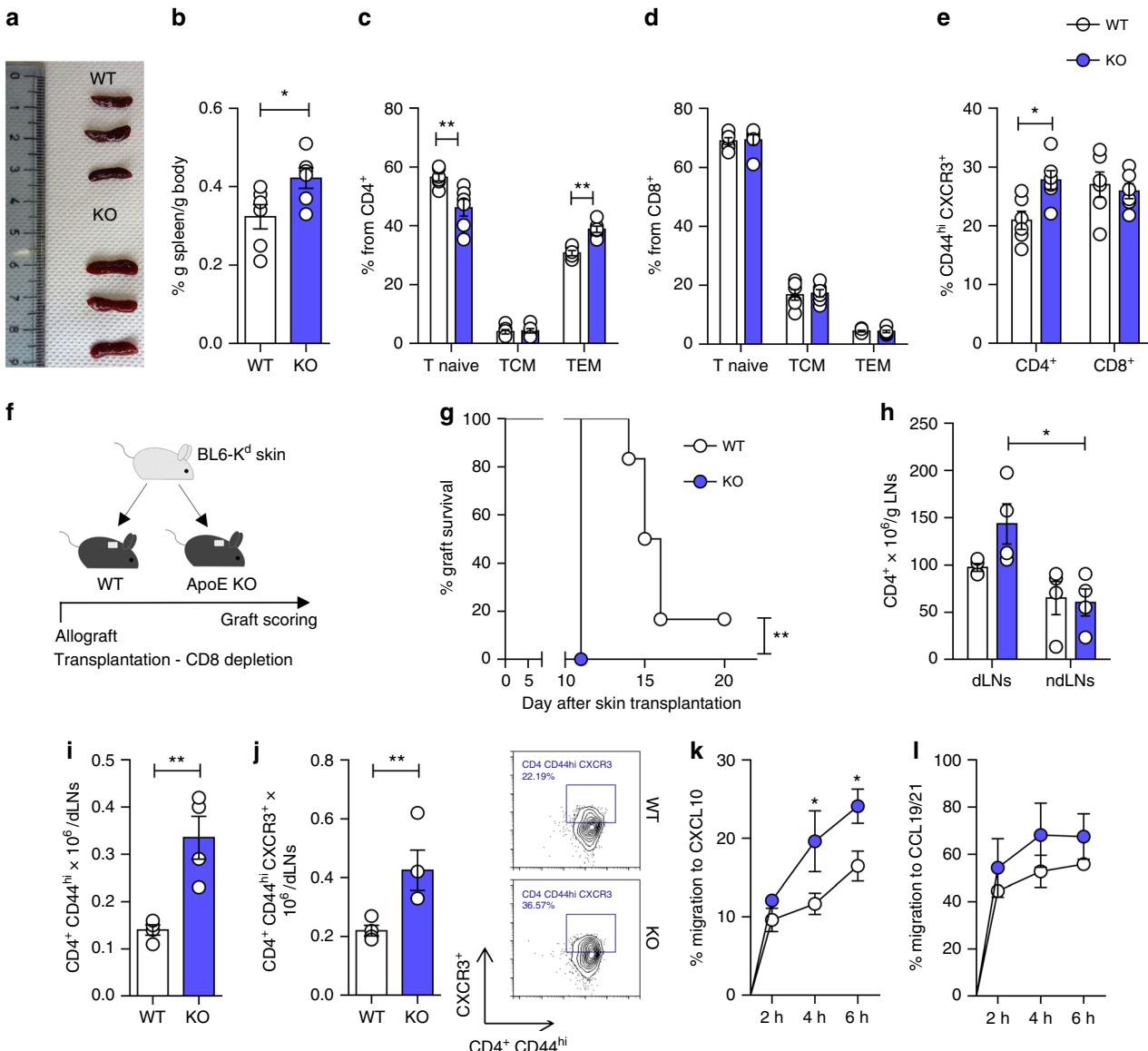

**Fig. 1** ApoE deficiency promotes CD4$^+$ T-cell activation and skin allograft rejection. **a–b** Representative images (**a**) and percentage of spleen weight corrected for body weight (**b**) from WT and apoE KO mice. **c–d** Frequency of CD4$^+$ (**c**) and CD8$^+$ (**d**) T-cells subsets in the spleen of WT and apoE KO mice. **e** Frequency of CD4$^+$ and CD8$^+$ activated CD44$^{hi}$ CXCR3$^+$ T cells in the spleen of WT and apoE KO mice. **f** Graphic representation of a typical skin allograft transplantation experiment: a piece of tail skin from C57BL/6K$^d$ donor was transplanted on the back of WT or apoE KO mice and graft survival was scored up to 3 weeks. **g** Graft survival following skin allotransplantation. Survival of less than 50% of the donor skin was recorded as rejection. **h** Number of CD4$^+$ T cells infiltrating the lymph nodes (axillary and brachial) draining (dLNs) and (contra-lateral inguinal) non-draining (ndLNs) the graft, corrected for LNs weight. **i–j** Number of activated CD4$^+$ T cells CD44$^{hi}$ (**i**) and CD44$^{hi}$ CXCR3$^+$ (**j**) in the dLNs after allograft rejection; representative dot plots are shown. **k–l** Migratory response of T lymphocytes isolated from draining lymph nodes, after allograft rejection, to the chemokines CXCL10 (**k**) and CCL19/21 (**l**) measured by transwell. $N = 4$ (**g–l**) or 6 (**a–e**) per group. Statistical analysis was performed with unpaired $T$-test (**b**, **i**, **j**), Gehan-Breslow-Wilcoxon test (**g**) and two-way Anova (**c–e**, **h**, **k**, **l**). Data are reported as mean ± SEM; *$p < 0.05$, **$p < 0.01$

from either apoE KO or WT mice (Fig. 2a). Rejection of allogeneic skin grafts by WT and apoE KO mice transplanted with apoE KO BM was significantly faster compared to that of WT BM recipients (Fig. 2b). Plasma cholesterol levels in WT mice transplanted with apoE KO BM were similar to those of apoE KO mice transplanted with WT BM, as was cholesterol distribution in the different lipoprotein subclasses (VLDL, LDL and HDL) (Fig. 2c, d and Supplementary Figure 4A), ruling out the possibility that faster rejection was simply the consequence of hypercholesterolemia. Accordingly, apoE KO mice, independently of the BM transplanted, displayed increased levels of CD4$^+$ T$_{EM}$, CD4$^+$CD25$^+$ and CD4$^+$CD44$^{hi}$ activated T cells (usually associated

with inflammation[17,38]) in the circulation as well as in draining and non-draining lymph nodes and in the spleen (Supplementary Figure 4B-G). This profile mirrored the immuno-inflammatory response associated to hypercholesterolemia and confirmed that allogenic skin graft rejection was independent of the systemic metabolic environment, suggesting a mechanism dependent on leukocyte-produced apoE. To validate this hypothesis, T cells isolated from peripheral lymph nodes of WT BM to WT mice, WT BM to apoE KO or from apoE KO BM to WT were challenged with splenocytes from B6K$^d$ donor mice. Alloreactive CD4$^+$Ki67$^+$ (proliferating) and polarized CD4$^+$T$_{EM}$ but not CD8$^+$Ki67$^+$ nor CD8$^+$T$_{EM}$ cells were significantly increased in

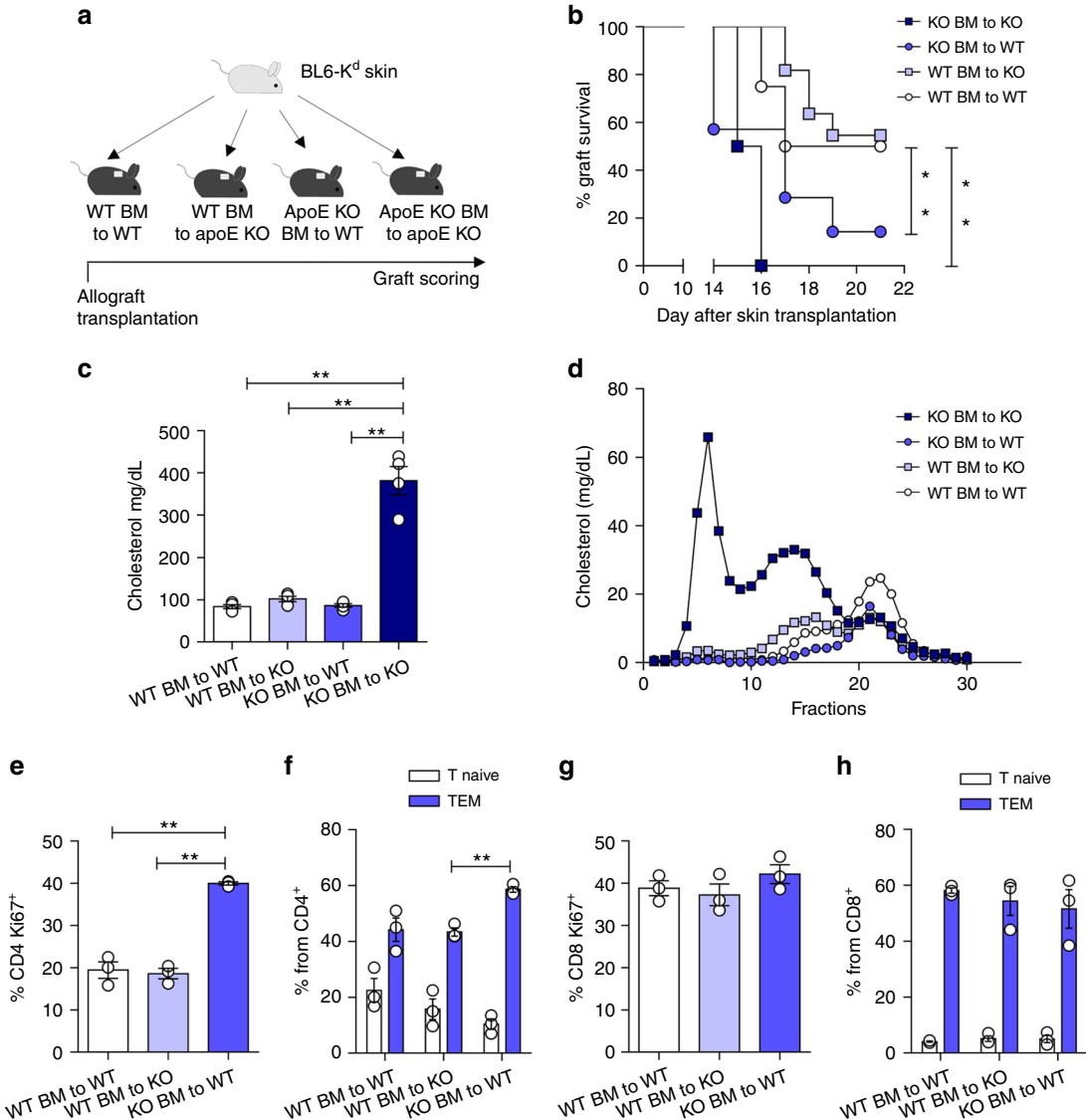

**Fig. 2** Hematopoietic deficiency of apoE promotes allograft rejection and allogenic T-cell expansion. **a** Graphic representation of skin allograft transplantation experiment: a piece of tail skin from C57BL/6K$^d$ donor was transplanted on the back of bone marrow transplanted (BMT) WT and apoE KO (BMT) mice; graft survival was scored up to 3 weeks. **b** Percentage of graft survival following skin allotransplantation in BMT mice. Survival of <50% of the donor skin was recorded as rejection. **c** Plasma cholesterol determination (mg/dL) after allograft rejection in BMT mice. **d** FPLC profiles of cholesterol distribution in the different lipoprotein subclasses (VLDL, LDL and HDL) from pooled plasma of grafted BMT mice ($n = 4$ for each group). **e** and **g** Proliferation of CD4$^+$ (**e**) and CD8$^+$ (**g**) lymphocytes isolated from draining lymph nodes of BMT-grafted mice, determined as Ki67$^+$ staining following challenge with splenocytes isolated from C57BL/6K$^d$ donor. **f** and **h** Polarization of CD4$^+$ (**f**) and CD8$^+$ (**h**) lymphocytes isolated from draining lymph nodes of BMT-grafted mice toward T$_{EM}$ subset after challenge with splenocytes isolated from C57BL/6K$^d$ donor. $N = 4$ per group. Statistical analysis was performed with Gehan-Breslow-Wilcoxon test (**b**) and two-way Anova (**c**, **e–h**). Data are reported as mean ± SEM; $*p < 0.05$, $**p < 0.01$

apoE KO BM to WT (those presenting a faster allograft rejection), compared to WT BM to apoE KO and WT BM to WT (Fig. 2e–h). These data suggest that rejection was driven by apoE deficiency in the hematopoietic compartment, irrespectively of the genetic background of the recipient, and also support the hypothesis that this effect might selectively apply to CD4$^+$ T-cell priming.

**DC-derived apoE affects the adaptive immune response**. To further define the cellular source of apoE involved in the immune phenotype of apoE KO mice, we analyzed its expression among different subsets of leukocytes compared to the liver. Substantial expression of apoE was observed not only in macrophages but

also in DCs (both spleen-derived DCs and bone marrow derived DCs, BMDCs) while it was almost undetectable in T cells (Fig. 3a, b). As DCs of both donor and recipient origin are responsible for the initiation of direct and indirect alloresponses, respectively[39], we investigated whether apoE deficiency impacted DC antigen presentation ability and induction of a primary immune response by T cells. An allogeneic mixed lymphocyte reaction (MLR) was setup using responder Balb/c T cells stimulated by WT or apoE KO C57BL/6 DCs. Spleen-derived DCs isolated from apoE KO mice were more efficient than those from WT mice at inducing BALB/c CD4$^+$ but not CD8$^+$ T cell proliferation (Fig. 3c–e). To further confirm these results in a nominal antigen-specific response, WT and apoE KO DCs were pulsed with peptides specific for OTI or OTII T cells and co-

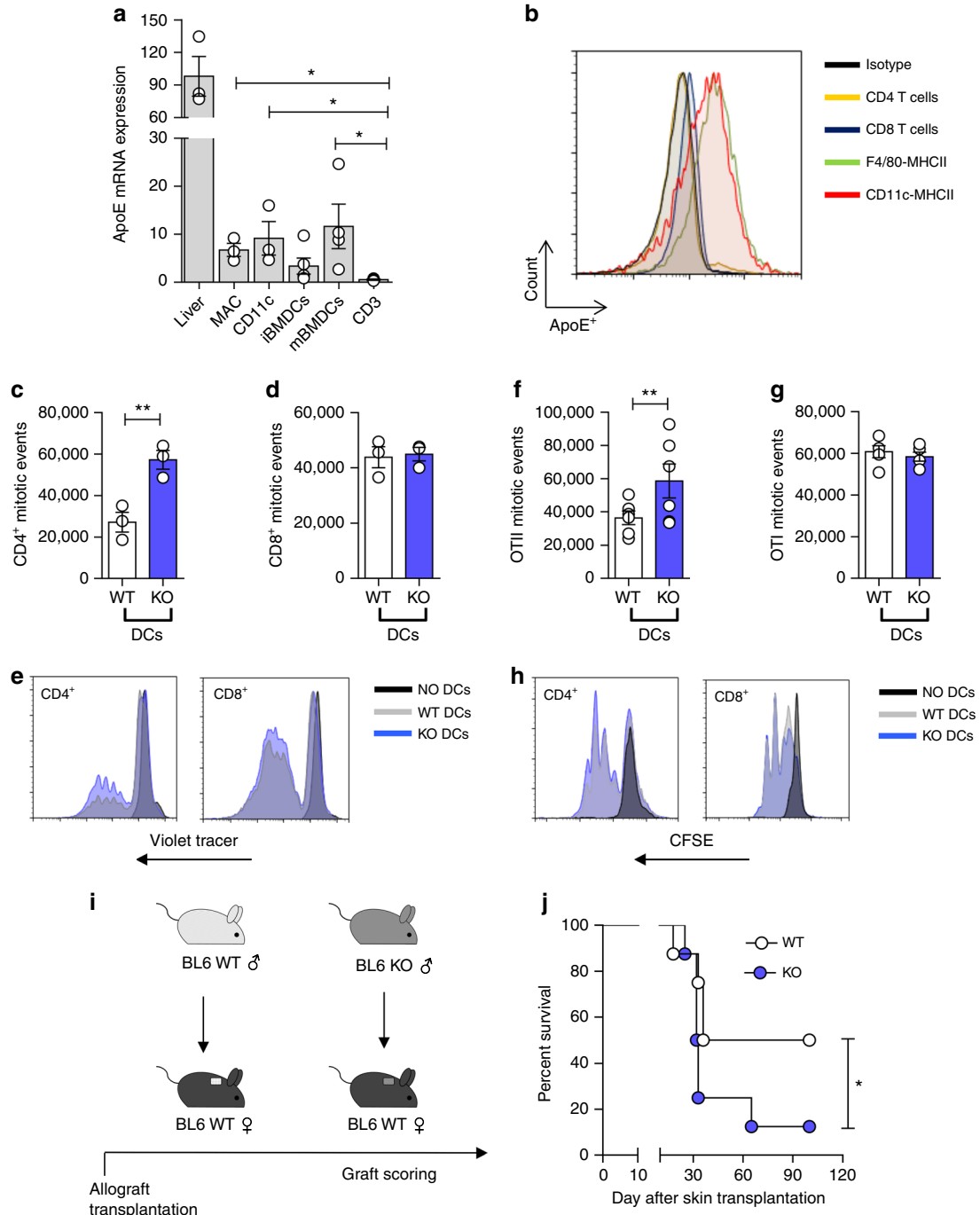

**Fig. 3** ApoE regulates DC activation. **a** mRNA expression relative to RPL (L Ribosomal Protein) of apoE in the liver, resident peritoneal macrophages (MAC), spleen-derived DCs (CD11c+), bone marrow derived DCs (immature iBMDCs and mature mBMDCs) and T cells (CD3+) of WT animals. **b** representative histogram showing apoE protein determination through flow cytometry in T cells (CD4+ and CD8+), macrophages (F4/80+ MHCII+) and DCs (CD11c+ MHCII+) of WT mice. **c–e** Proliferation of allogenic BALB/c CD4+ (**c**) and CD8+ (**d**) T cells with spleen-derived DCs from WT and apoE KO C57BL/6 mice; representative histograms are presented in **e**. **f–h** Proliferation of transgenic OTII CD4+ (**f**) and OTI CD8+ (**g**) T cells with BMDCs isolated from WT and apoE KO mice and pulsed with OTII or OTI peptide; representative histograms are presented in **h**. **i** Graphic representation of skin allograft transplantation: a piece of tail skin from a C57BL/6 male donor either WT or apoE KO was transplanted on the back of WT female mice; graft survival was scored up to 100 days. **j** Percentage of graft survival following skin transplantation. Graft survival of <50% of the donor skin was recorded as rejection. $N = 3$ (in triplicates **c–d**), $N = 4$ (**a**), $N = 6$ (**f–g**), $N = 8$ (**j**) per group. Statistical analysis was performed with unpaired *T*-test (**a**, **c**, **d**, **f**, **g**), Gehan-Breslow-Wilcoxon test (**j**). Data are reported as mean ± SEM; *$p < 0.05$, **$p < 0.01$.

cultured with OTI or OTII TCR-transgenic T cells (antigen specific, non-mismatch-based assay). A significantly increased proliferation was observed in OTII T cells incubated with apoE KO-pulsed DCs compared to WT-pulsed DCs, while OTI T-cell proliferation was similar irrespective to the DC origin (Fig. 3f–h).

Moreover, OTII T cells incubated with apoE KO DCs displayed increased production of IFNγ compared to those incubated with WT DCs (Supplementary Figure 5A, B). In contrast, T cells isolated from apoE KO mice underwent a similar rate of proliferation compared to WT T cells when co-cultured with BALB/c

spleen-derived DCs (Supplementary Figure 5C). In line with these results, conjugates between BMDCs of BALB/c mice and T cells from WT and apoE KO mice were similar in both naïve and activated T cells, ruling out differences in cell:cell interactions due to apoE deficiency in T cells (Supplementary Figure 5D). Finally, phosphorylation of Akt, a downstream target of TCR/CD28 co-stimulation, was comparable in T cells isolated from WT and apoE KO mice (Supplementary Figure 5E). These results excluded the possibility that apoE deficiency directly affects T-cell activation and proliferation while indicating a selective effect on DCs. To test this hypothesis, we transplanted male WT and apoE KO skin (a source of tissue-resident DCs) to female WT recipients (Fig. 3i). This system was chosen as rejection depends on the presentation of endogenously expressed, male-specific minor histocompatibility antigen H-Y[40,41], hence does not involve host-derived female DCs as these do not express the endogenous HY antigen. As shown in Fig. 3j, apoE deficiency in transplanted skin significantly reduced graft survival, indicating that lack of apoE expression by graft-derived antigen presenting cells might be responsible for enhanced allograft rejection.

**ApoE deficiency affects DC antigen presentation function.** The data presented thus far indicate that apoE deficiency leads to proliferation and preferential differentiation of CD4$^+$ T cells toward an effector phenotype, an effect dependent on DC activity. Therefore, we sought to investigate the mechanism by which apoE regulates DC antigen presenting function. An analysis of DC distribution showed a significant increase of CD11c$^+$ DCs in the spleen of apoE KO mice as compared to their WT littermates (Fig. 4a, b), in line to what has been previously reported[42]. Moreover, we demonstrated that within the DC population, the conventional DC2 subset (cDC2: CD11c$^+$MHCII$^+$B220$^-$CD11b$^+$CD8$^-$) was significantly increased in apoE KO compared to WT mice (Fig. 4c), while no difference was observed in the cDC1 subset (cDC1: CD11c$^+$MHCII$^+$B220$^+$CD11b$^-$CD8$^+$) (Fig. 4c) or in plasmacytoid DCs (CD11c$^+$MHCII$^+$B220$^+$) (Supplementary Figure 6A). Furthermore, MHC class II molecule (MHC-II) surface expression was significantly increased in cDC2 (the subset which more efficiently activate CD4$^+$ T cells) from apoE KO compared to WT mice (Fig. 4d and Supplementary Figure 6L), while similar expression was observed on cDC1 cells (Fig. 4d) which specialize antigen cross-presentation to CD8$^+$ T cells. Of note, MHC class I molecule (MHC-I) expression was similar in both cDC1 and cDC2 subsets (Fig. 4e). Likewise, the distinct phenotype of apoE KO cDC2s was restricted to MHC-II molecules as the expression of co-stimulatory molecules CD40, CD86 and CD80 was comparable to WT DCs (Supplementary Figure 6B-D). Consistent with this profile, BMDCs derived from WT and apoE KO mice presented a similar expression of CD86 and CD80 and production of IFNγ, IL-12, IL-23 and IL-10 following stimulation with LPS (Supplementary Figure 6E-I). These findings suggest that apoE deficiency might be associated with increased antigen presentation, but not co-stimulatory function. To validate this hypothesis, we tested in vitro and in vivo antigen uptake and presentation using the Eα green fluorescence protein (EαGFP)/YAe, a system which allows tracking of antigen uptake/processing and presentation in situ[43,44]. In vitro, BMDCs from apoE KO mice displayed a significant increase in antigen uptake (GFP positivity) and presentation (Eα:IAb complex positivity) compared to WT (Fig. 4f). In vivo, after peritoneal injection of the EαGFP antigen, DCs from apoE KO mice showed increased uptake of EαGFP as compared to WT (Fig. 4g); in addition, GFP$^+$DCs from apoE KO mice also displayed higher levels of MHC-II expression (Fig. 4h). These observations indicate that apoE influences antigen processing and presentation by DCs.

**ApoE impacts DC cholesterol homeostasis.** We next interrogated the molecular mechanisms responsible for this effect. It has been reported that key receptors involved in hematopoietic stem progenitors cell proliferation are enriched in lipid rafts and their expression can be modulated by apoE upon cholesterol mobilization from the membrane[21]. We therefore asked whether this mechanism could also be operational in DCs and performed detailed lipidomic analyses of WT and apoE KO spleen-derived DCs. ApoE KO DCs displayed a significant increase in cholesterol and oxysterols, including 25-hydroxycholesterol (25-OH), 7β-hydroxycholesterol (7β-OH) and 7-Keto-cholesterol (7-KETO) levels, concomitant to a decrease in the levels of several phospholipids (Fig. 5a, Supplementary Figures 7A and 8A, B). The accumulation of cellular cholesterol in the DCs from apoE KO mice was further confirmed by filipin staining (Fig. 5b). As a consequence of sterols accumulation, apoE KO DCs displayed a decreased expression of genes involved in cholesterol and fatty acid uptake (LDLR, CD36) and cholesterol synthesis (HMGCoA reductase and 24-Dehydrocholesterol Reductase, DHCR24) (Fig. 5c), while the protein levels of cholesterol 25-hydroxylase (CH25H), the enzyme that catalyzes cholesterol conversion to 25-OH, was significantly higher in apoE KO DCs (Fig. 5d). Of note, no differences were observed between WT and apoE KO DCs in the expression of genes promoting cholesterol efflux (ABCA1 and ABCG1) (Fig. 5c) as well as in lipid raft content and MHC-II expression after LXR activation in WT and apoE KO BMDCs (Supplementary Figure 7C, D); however, this is not surprising as the level of the dominant LXR ligand[45], namely desmosterol, was slightly decreased in DCs from apoE KO compared to those from WT mice (Fig. 5a, Supplementary Figure 7A). This is consistent with reduced protein levels of 7-dehydrocholesterol reductase (DHCR7, Supplementary Figure 7B), an enzyme involved in desmosterol synthesis, detected in apoE KO compared to WT DCs, thus perhaps reflecting a negative feedback on the cholesterol biosynthetic pathway. Importantly, increased cholesterol accumulation led to increased lipid rafts in apoE KO compared to WT DCs (Fig. 5e). Accordingly, membrane expression of toll-like receptor 4 (TLR4), a key receptor involved in DC signal transduction located in rafts, was significantly increased in apoE KO DCs compared to WT (Fig. 5f); this effect, however, was unrelated to the LPS responsiveness which was similar in WT and apoE KO DCs (Supplementary Figure 6). We therefore investigated whether membrane cholesterol accumulation also affected the distribution of receptors involved in antigen presentation, such as MHC-II molecules, known to localize in lipid rafts[46]. We observed an increased localization of MHC-II molecules in lipid raft microdomains in apoE KO DCs as compared to WT (Fig. 5g, h), further supporting the concept that apoE, through the regulation of membrane lipid homeostasis, plays a physiological role in controlling the surface distribution of MHC-II molecules, thus finely tuning the modality of DC antigen presentation to CD4$^+$ T lymphocytes.

**Monocyte-DC function is differently affected by human apoE.** To assess the effect of apoE in human DCs, we took advantage of the presence of three common isoforms of apoE in humans namely apoE2, apoE3, and apoE4 (with ε2, ε3, and ε4 allele frequencies of about 7, 78, and 14%, respectively)[47]. ApoE3 is considered the parent form and contains a Cysteine (Cys) at positions 112 and an Arginine (Arg) at position 158, while apoE2 and apoE4 isoforms differ by amino acid substitutions of a Cys at position 158 and Arg at position 112, respectively, causing structural alterations which impact protein function[48]. As a consequence, lipid-binding properties differ between isoforms and result in higher affinity of apoE4 for lipid particles, thus

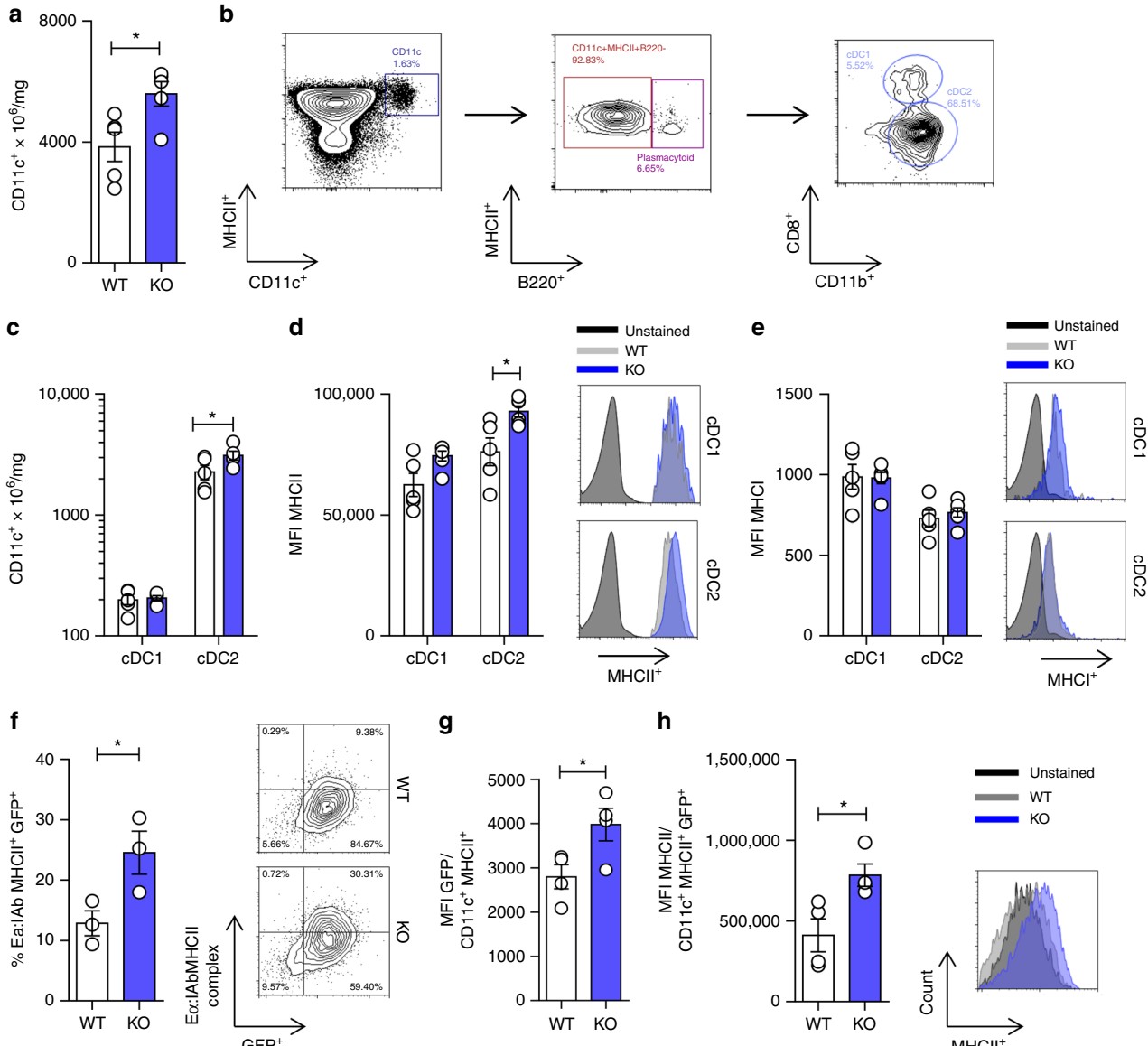

**Fig. 4** ApoE deficiency alters the phenotype of spleen-derived DCs. **a** Number of CD11c$^+$ from WT and apoE KO spleen normalized by spleen weight. **b** Gating strategy for DC subsets phenotyping. **c** Number of cDC1 (CD11c$^+$MHCII$^+$B220$^-$CD11b$^-$CD8$^+$) and cDC2 (CD11c$^+$MHCII$^+$B220$^-$CD11b$^+$CD8$^-$) corrected for spleen weight of WT or apoE KO mice. **d** Median fluorescence intensity (MFI) of MHCII in cDC1s and cDC2s from WT or apoE KO mice; representative histograms are shown. **e** Median fluorescence intensity (MFI) of MHCI in cDC1s and cDC2s from WT or apoE KO mice; representative histograms are shown. **f** Percentage of double positive GFP$^+$/Eα:I MHCII$^+$ BMDCs after in vitro antigen uptake and presentation assay with Eα peptide; representative contour plots are shown. **g–h** Median fluorescence intensity (MFI) of GFP (**h**) and MHCII (**i**) in DCs after in vivo antigen uptake and presentation assay with Eα peptide; representative contour plots and histograms are presented. $N = 3$ (**f**), $N = 4$ (**g–h**), $N = 5$ (**a–e**) per group. Statistical analysis was performed with unpaired $T$-test. Data are reported as mean ± SEM; *$p < 0.05$

causing a better interaction with VLDL, but less efficient with HDL compared to apoE3 and apoE2 isoforms[49]. We performed an immuno-metabolic phenotyping of carriers of apoE isoforms identified in the PLIC study[50,51] investigating both biochemical parameters and immune signature. Plasma total and LDL cholesterol profiles differed in apoE2 (ε2/ε3), apoE3 (ε3/ε3) and apoE4 (ε4/ε3) carriers (Fig. 6a, b) while plasma levels of triglycerides were similar (Fig. 6c and Supplementary Table 1). Next, we investigated by flow cytometry whether functional alterations of apoE in carriers of the apoE4 isoform reflected a different CD4$^+$ T-cell subset distribution. ApoE4 carriers presented increased levels of circulating CD4$^+$ T$_{EM}$, while CD4$^+$ T$_{naive}$, CD4$^+$ T$_{CM}$ and Treg cells were similar to those of apoE3 carriers (Fig. 6d–i and Supplementary Figure 9A). As in mice the

expansion of T$_{EM}$ cells was independent of plasma cholesterol, we evaluated whether this phenotype in apoE4 carriers might arise as a result of enhanced DC antigen presentation ability, by performing a mixed lymphocyte reaction (MLR) between monocyte-derived DCs (MDC) pulsed with LPS and naive CD4$^+$ T cells in autologous serum (Fig. 6j). With this experimental setting, apoE is produced by DCs and also carried by subject's lipoproteins. ApoE4 MDCs induced the differentiation of significantly more CD4$^+$ T$_{naive}$ cells toward the CD4$^+$ T$_{EM}$ phenotype when compared to MDCs from apoE3 and apoE2 carriers, while the proportion of CD4$^+$ T$_{CM}$ cells was similar in all carriers (Fig. 6k, Supplementary Figure 9B, C). We also excluded a direct effect of apoE isoform on T cells by performing MLR in the presence of CD4$^+$ T$_{naive}$ cells from a donor expressing the parent apoE3

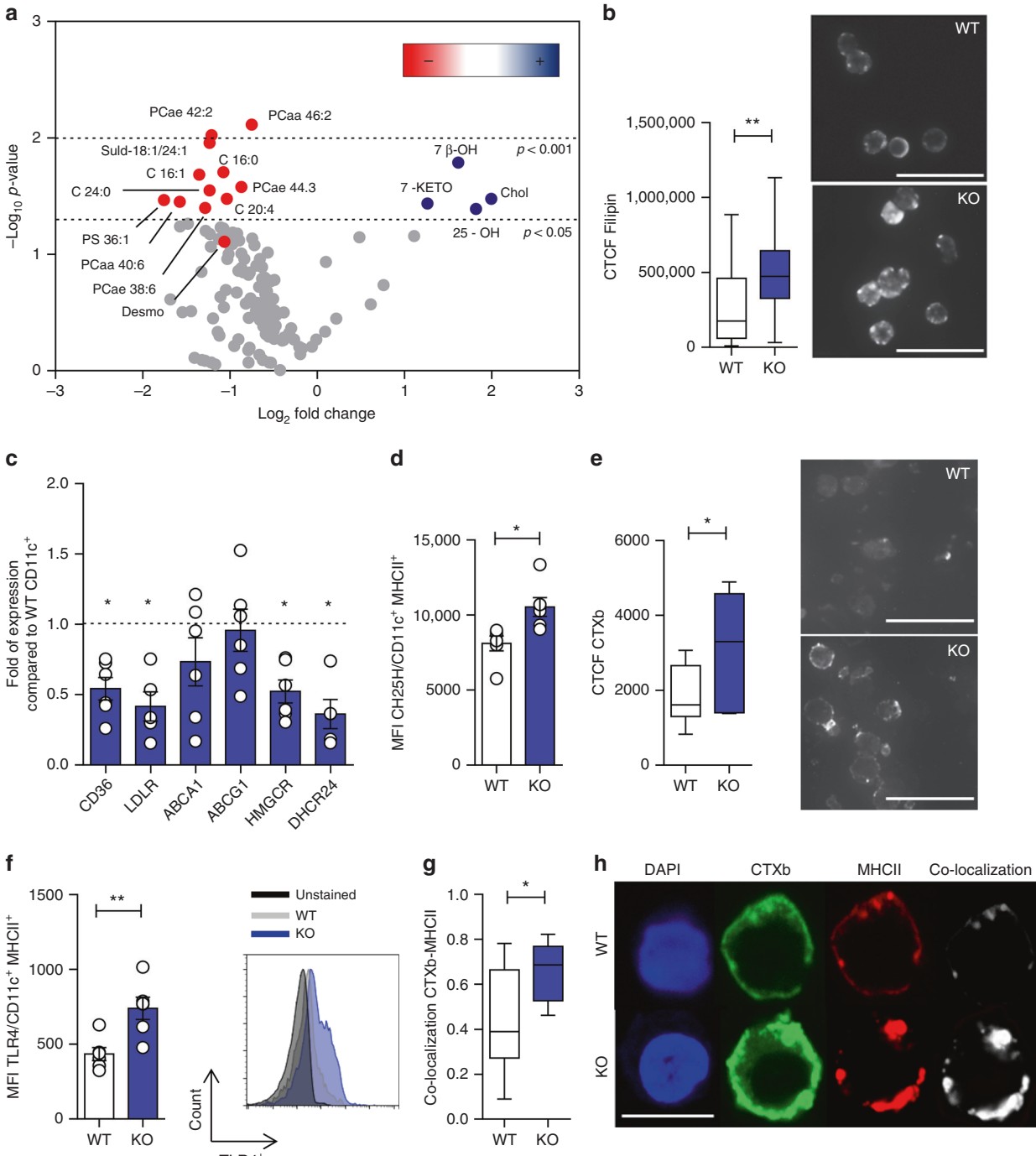

**Fig. 5** ApoE deficiency affects lipid composition of DCs. **a** Volcano plot from extensive lipidomic analysis (fatty acids, phospholipids and oxysterols) in purified CD11c+ DCs isolated from the spleen of WT or apoE KO mice. The volcano plot correlates fold-change expression (expressed as log2) and significance between two groups (WT vs. apoE KO), using a scatter plot view. The y-axis is the negative log10 of p-values (a higher value indicates greater significance as indicated by dashed lines) and the x-axis is the difference in levels of metabolites between two experimental groups. Metabolites which are significantly increased are shown by blue dots while red dots represent those decreased. **b** Total cell fluorescence (CTCF) of free cholesterol (filipin staining) in purified CD11c+ DCs isolated from the spleen of WT or apoE KO mice, calculated with ImageJ software; representative pictures are shown (×20 magnification, scale bar 25 μm). **c** Expression of CD36, LDLR, ABCA1, ABCG1, HMGCR and DHCR24 mRNA by purified CD11c+ DCs isolated from the spleen of WT or apoE KO mice. **d** Mean fluorescence intensity (MFI) of CH25H in CD11c+ MHCII+ DCs of WT and apoE KO determined by flow cytometry. **e** Total cell fluorescence (CTCF) of lipid rafts (CTXb) in purified CD11c+ DCs isolated from the spleen of WT or apoE KO mice, calculated with ImageJ software; representative pictures are presented (×20 magnification, scale bar 25 μm). **f** Median fluorescence intensity (MFI) of TLR4 expression in CD11c+ MHCII+ DCs of WT and apoE KO determined by flow cytometry; representative histograms are shown. **g** and **h** Pearson's correlation coefficient between lipid rafts (CTXb) and MHCII calculated with ImageJ software (**g**) in purified CD11c+ DCs isolated from the spleen of WT or apoE KO mice; representative pictures from confocal microscopy staining for CTXb (green), MHCII (red) and DAPI (nucleus, blu) are presented in **h** (×63 magnification, scale bar 10 μm). $N = 4$ (**a**, **b**, **e**) and $N = 6$ (**c**, **d**, **f**–**h**) per group. Statistical analysis was performed with unpaired T-test. Data are reported as mean ± SEM; box and whiskers shown min to max values; *$p < 0.05$, **$p < 0.01$

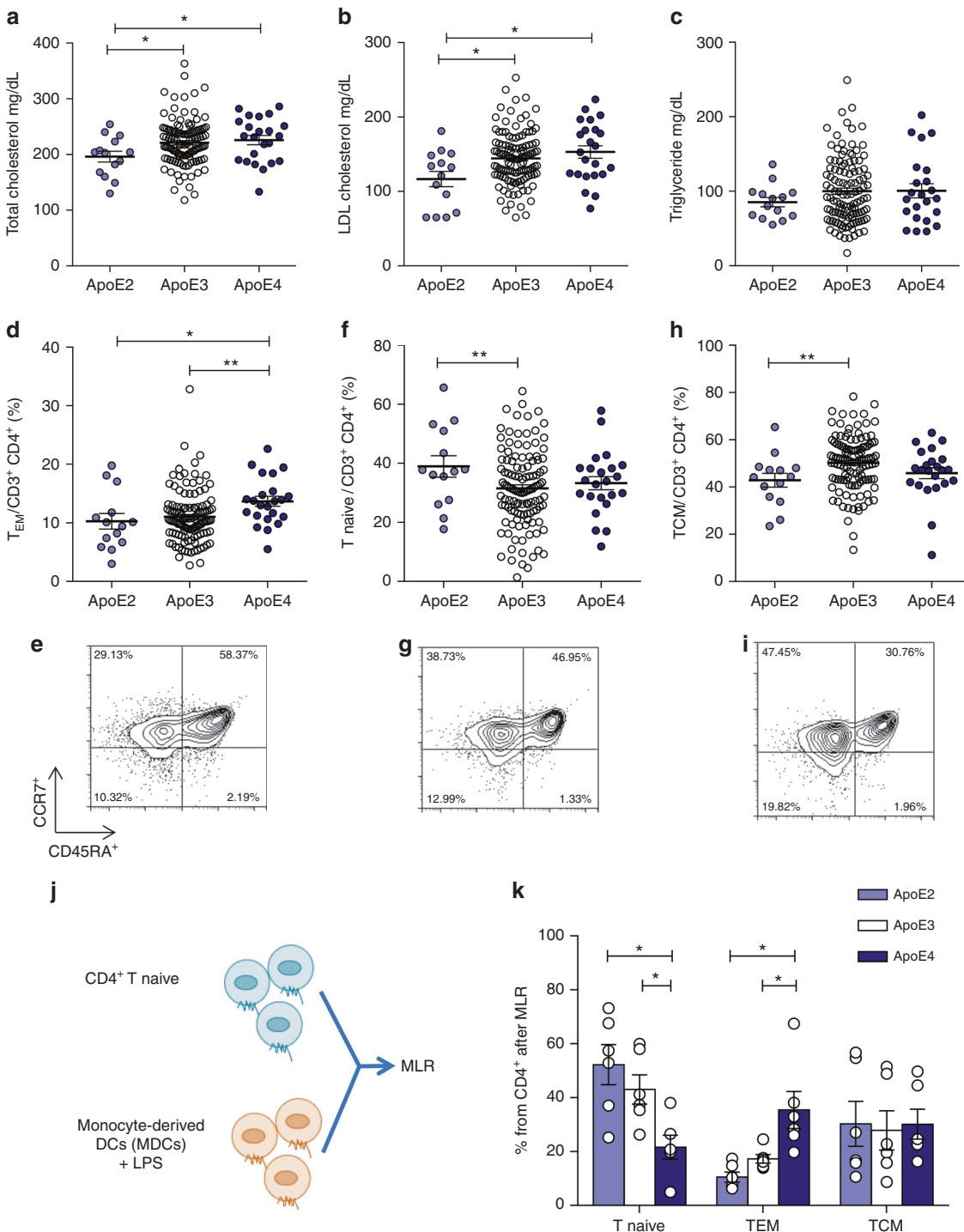

**Fig. 6** Human carriers of apoE4 isoform present a more activated immune profile as compared to carriers of the wild-type apoE3 isoform. **a–c** Plasma levels of total cholesterol (**a**), LDL-cholesterol (**b**) and triglycerides (**c**) from human carriers of apoE isoforms. **d–i** Percentage of circulating $T_{EM}$ (CD45RA-CCR7lo) (**d**), $T_{naive}$ (CD45RA + CCR7hi) (**f**) and $T_{CM}$ (CD45RA-CCR7hi) (**h**) CD4$^+$ in carriers of the different apoE isoforms; representative dot plots are reported (**e**, **g**, **i**). **j** Graphic representation of mixed lymphocyte reaction (MLR) by CD4$^+$ T naïve cells stimulated with monocyte-derived DCs (MDCs) pulsed with LPS (1 μg/mL for 2 days) from human carriers of apoE isoforms. **k** Polarization of CD4$^+$ T cells after MLR between mature MDCs and syngenic T naive cells. $N = 6$ per group (**k**). Statistical analysis was performed with two-way Anova. Data are reported as mean ± SEM; *$p < 0.05$, **$p < 0.01$

isoform, thus confirming that incubation with apoE4 MDCs results in increased CD4$^+$ $T_{EM}$ differentiation (Supplementary Figure 10A). Given the immunomodulatory role of HDL[52], we asked whether the decreased affinity of apoE4 for HDL might affect their ability to promote cholesterol efflux, thus causing an alteration in intracellular cholesterol handling leading finally to a polarization toward a pro-inflammatory immune phenotype.

Cellular cholesterol levels were indeed significantly increased in apoE4 MDCs compared to apoE3 and apoE2 MDCs (Supplementary Figure 10B, C). We therefore performed a comprehensive characterization of MDCs from apoE4 carriers compared to apoE3 MDCs, as apoE2 and apoE3 isoforms have similar affinity for HDL. We observed that treatment of MDCs with LPS increased lipid rafts enrichment in apoE4 carriers compared to

apoE3 carriers (Fig. 7a). In addition, the expression of MHC-II molecule HLA-DR, but not CD80, was significantly increased in apoE4 compared to apoE3 MDCs, regardless of LPS stimulation (Fig. 7b, c). These data confirmed that human carriers of the apoE4 isoform with reduced ability to mobilize cholesterol display increased CD4$^+$ T$_{EM}$ biased differentiation as a consequence of enhanced antigen presentation by DCs.

**Restoration of cholesterol efflux improves apoE4 DC function.** In animal models, cholesterol efflux mediated by apoE containing-nascent HDL from hematopoietic stem cell progenitors has been shown to reduce membrane lipid rafts[21], thus dramatically affecting DC antigen presentation[46]. Therefore, we explored whether the use of serum containing apoE3-HDL could rescue the phenotype of apoE4 MDCs by culturing apoE4 DCs

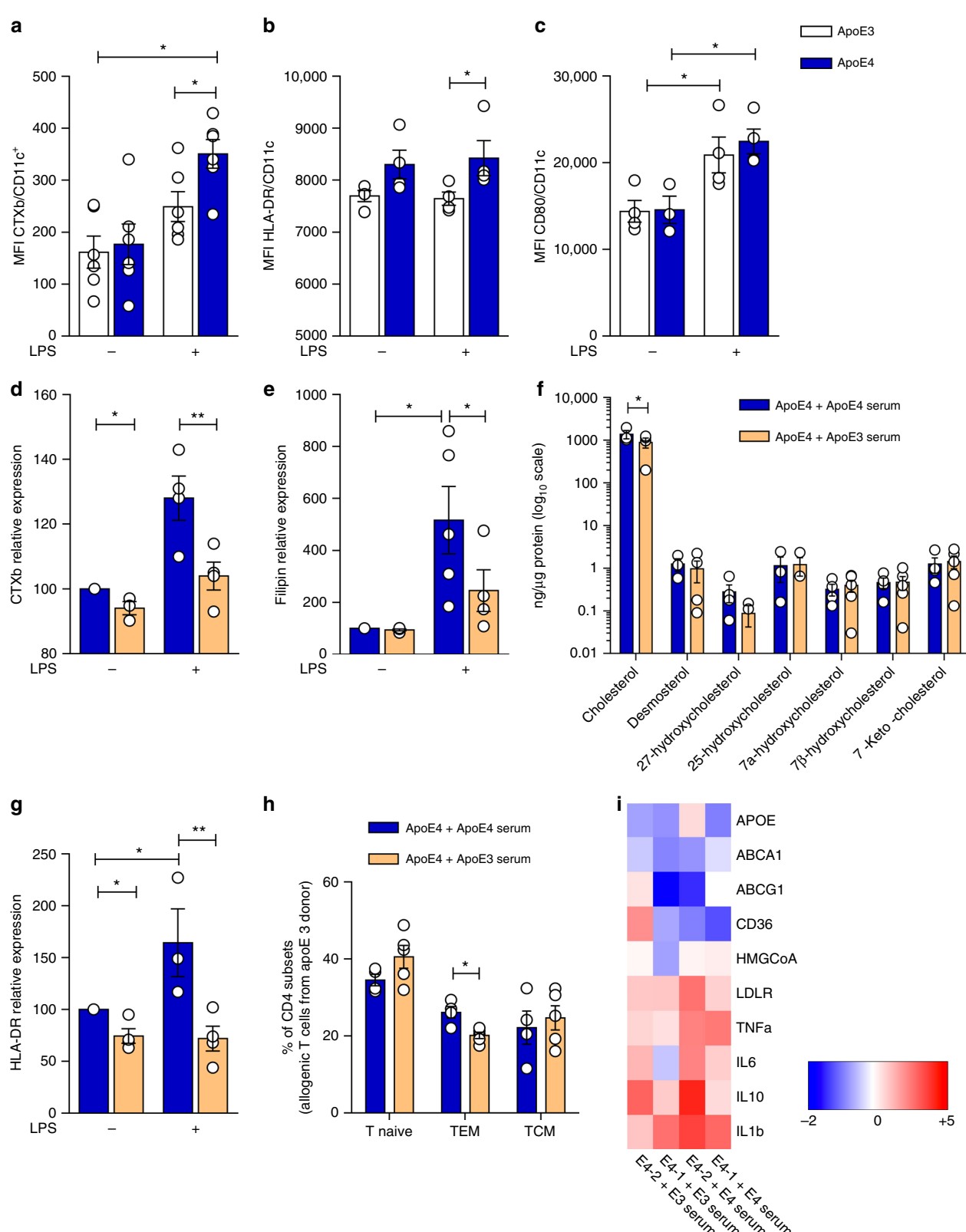

with medium supplemented with serum from apoE4 and apoE3 carriers. ApoE3 serum reduced the enrichment of lipid rafts (Fig. 7d) and cholesterol (Fig. 7e, f) in the membrane of apoE4 MDCs, especially after LPS-induced maturation. Reduced lipid rafts and cholesterol accumulation in apoE4 MDCs incubated with a serum containing apoE3-HDL was associated with decreased expression of HLA-DR (Fig. 7g), which resulted in decreased CD4+ T_EM development (Fig. 7h) compared to cells treated with apoE4 serum. In addition, preliminary data indicate that this approach might also affect the expression of cholesterol-related and inflammatory genes (Fig. 7i), although experiments in a larger number of carriers of different apoE isoforms will be required to validate this finding. Collectively, these data suggest that exogenously added apoE3 can prevent the pro-inflammatory T-cell phenotype induced by human DCs from apoE4 carriers.

## Discussion

Apolipoprotein E (ApoE) is a 34.2 kDa glycosylated protein, mainly bound to plasma lipoproteins but also present on immune cell membrane[20], which plays a crucial role in cellular and systemic cholesterol metabolism.

Here we show that apoE deficiency enhances T-cell activation as a result of increased DC antigen presenting function, as a consequence of lipid raft enrichment and enhanced MHC-II molecule clustering on the DC membrane. In antigen presenting cells (APCs), lipid rafts act as cholesterol-enriched specialized microdomains where MHC-II molecules concentrate, optimizing T-cell activation[46,53]. Lack of DC-produced apoE results in impaired removal of cholesterol from the membrane and lipid raft accumulation. As a consequence, apoE KO mice display enhanced expansion of pro-inflammatory CD4+ T cells (CD44hi and T_EM) and enhanced skin allograft rejection. We show that this response is independent of dyslipidemia—a well-known factor promoting systemic inflammation, and previously hypothesized to impact T-cell activity[54,55] but rather the consequence of apoE deficiency in bone marrow-derived antigen-presenting cells. ApoE was previously shown to affect macrophage function[56]. Here we show that apoE is also produced by mature DCs and its deficiency in myeloid cells recapitulates the phenotype observed in apoE KO mice.

Cellular lipid homeostasis and inflammatory responses are reciprocally regulated. Loss of this balance—as it occurs in macrophages lacking the ABC transporters ABCA1 and ABCG1 —leads to accumulation of free cholesterol in plasma membrane resulting in increased TLR cell surface concentration and enhanced inflammatory responses to LPS[57,58]. Cholesterol accumulation in cells activates LXR, which promotes its efflux to lipid-poor apoA-I and favors the reverse transport of cholesterol to the liver for elimination[59]. Furthermore, LXR deficiency results in increased inflammation[60], aberrant lymphocyte proliferation[15] and impaired antigen presentation by B cells and DCs[5]. The anti-inflammatory effects of LXR were recently proposed, at least in macrophages, to be the consequence of ABCA1-dependent changes in membrane lipid organization but not of LXR dependent ABCG1 or apoE modulation[61]. However, LXR-dependent repression of inflammatory gene expression in activated macrophages was also shown to be independent of cholesterol modulation and to rely on the inhibition of multiple NF-κB target genes (COX-2, IκBα, IL-1β, MCP-1, IL-6 and IL-1-receptor antagonist) after LPS, TNF-α, or IL-1β stimulation[62]. In line with this report, our observation that the expression of two key target genes of LXR (such as ABCA1 or ABCG1) is not increased in DCs from apoE KO, broadens the scenario by demonstrating the autocrine/paracrine role of apoE in controlling cellular cholesterol homeostasis in DCs beyond the role of LXR. A neutral effect on LXR pathway in apoE KO DCs might result from the balance between increased oxysterols levels and decreased levels of desmosterol, the final precursor of cholesterol[63] and the dominant physiological activator of LXR[45,64].

Our data indicate that apoE controls DC function through a mechanism which appears to be restricted to MHC-II molecule compartmentalization on the cell surface. In B cells, antigen uptake was shown to be dependent on lipid raft-mediated endocytosis, and, more importantly, localization and internalization of MHC-I and MHC-II molecules in different membrane domains, with MHC-II molecules showing a strong co-clustering with lipid rafts[65]. Our observations expand these findings to DCs showing that accumulation of lipid rafts due to apoE deficiency leads to an increased EαGFP peptide uptake and processing through MHC-II molecules, which specifically elicits a CD4+ but not CD8+ T-cell activation. Consistently, apoE deficiency in DCs boosted CD4+ but not CD8+ T-cell proliferation, further confirming that the modulation of cellular cholesterol by apoE impacts mainly MHC-II function. Further, in vitro inhibition of cholesterol biosynthesis perturbed raft composition resulting in the alteration of antigen processing and presentation to CD4+ T cells[66], which depends on MHC-II peptide complex clustering in DC lipid microdomains[67].

Finally, we show the translational relevance of our findings by demonstrating that DCs from human carriers of the apoE4 isoform present increased lipid rafts content. This might be the consequence of a decreased ability of apoE4 isoform to mobilize cholesterol from DCs compared to the WT apoE3 isoform, as previously demonstrated in macrophages[68]. This, in turn, leads to an enhanced antigen presentation mediated by HLA-DR, as well as to increased polarization of CD4+ naive T cells toward the pro-inflammatory effector memory phenotype when incubated with mature DCs.

Previously, increased lipid rafts accumulation has also been observed in monocytes of apoE4 carriers with severe sepsis and has been associated with higher cytokine levels upon ex vivo stimulation with TLR2, TLR4, or TLR5 ligands and increased organ injury as compared to apoE3 carriers[69] further supporting the relevance of apoE in humans. Altogether, these findings point

**Fig. 7** Lowering cellular cholesterol rescues the phenotype of apoE4 MDCs. **a–c** Median fluorescence intensity (MFI) of lipid rafts (CTXb, **a**), HLA-DR (**b**), CD80 (**c**) in immature (LPS−) and mature (LPS+) MDCs. **d** and **e** Relative quantification of lipid rafts (CTXb, **d**) and free cholesterol (filipin, **e**) in immature (LPS−) or mature (LPS+) apoE4 MDCs cultured with serum derived from an allogenic apoE4 donor as compared to an apoE3 donor. Data are presented as relative expression compared to immature apoE4 MDCs cultured with serum from an allogenic apoE4 donor. **f** Determination (ng/μg) of sterols and oxysterols by gas chromatography-mass spectrometry of mature (LPS+) apoE4 MDCs cultured with serum from an allogenic apoE4 or apoE3 donor. **g** Relative expression of HLA-DR in immature (LPS−) or mature (LPS+) apoE4 MDCs cultured with serum derived from an apoE4 donor as compared to immature (LPS−) apoE4 MDCs cultured with serum from an apoE4 donor; data are presented as relative expression (calculated from HLA-DR+ cells) compared to immature apoE4 MDCs cultured with serum from an apoE4 donor. **h** Differentiation of CD4+ T naive cells from a carrier of apoE3 isoform with allogenic mature (LPS+) apoE4 MDCs cultured with serum from an apoE4 or apoE3 donor. **i** Heat-map of lipid-related and inflammatory genes in mature (LPS+) MDCs from apoE4 carriers cultured in the presence of serum from allogenic apoE4 or an apoE3 donor; data are presented as relative to mean expression in apoE3 DCs (log2 scale). N = 4–6 per group. Statistical analysis was performed with two-way Anova. Data are reported as mean ± SEM; *p < 0.05, **p < 0.01

to a critical role for apoE in immune responses by supporting cholesterol and lipid mobilization from cells to HDL, which have been proposed to modulate the immune response[52].

In summary, our study provides compelling evidence for a physiological role of apoE in adaptive immune response through the modulation of membrane cholesterol levels and lipid rafts content in DCs. The restoration of cholesterol homeostasis following the administration of WT apoE3 rescues the activated phenotype observed in apoE4 DCs by decreasing lipid rafts and MHC-II abundance. These findings pave the way for the therapeutic exploitation of apoE as a tool to modulate immune responses.

## Methods

**Mice**. All mice used in this study were between the ages of 8–12 weeks. Male and female WT and apoE KO on the C57BL/6 background and BALB/c mice were purchased from Charles River (UK and Italy). Heterozygous (apoE+/−) were crossed and littermates apoE KO and WT were used. B6K$^d$ mice, which express H-2K$^d$ molecules as a transgene on the C57BL/6 background[70], Marylin female mice [71] and OTI and OTII transgenic mice were bred in house. Mice were housed four per cages and kept in a temperature-controlled environment (20 ± 2 °C, 50 ± 5% relative humidity) with a 12-hour light/dark cycle in an air-conditioned room and free access to food and water. The investigation conforms to the European Commission Directive 2010/63/EU and was approved by the Ethical Committee (Progetto di Ricerca 2012/02, Autorizzazione Ministeriale 1094/2016) and Home Office guidelines (PPL 70/7443) following approval by Queen Mary University of London Ethics Committee.

**Human samples**. The Progressione della Lesione Intimale Carotidea (PLIC) Study (a sub-study of the CHECK study) is a large survey of the general population of the northern area of Milan ($n = 2606$)[50,51], followed at the Center for the Study of Atherosclerosis, Bassini Hospital (Cinisello Balsamo, Milan, Italy). The Study was approved by the Scientific Committee of the Università degli Studi di Milano ("Cholesterol and Health: Education, Control and Knwoledge—Studio CHECK (SEFAP/Pr.0003)—reference number Fa-04-Feb-01) on February 4th, 2001. An informed consent was obtained by participants of the study in accordance with the Declaration of Helsinki.

Genomic DNA was extracted using Flexigene DNA kit (Qiagen, Milan, Italy) as previously described[51]. Genotyping for the apoE isoforms was performed by TaqMan allelic discrimination. Information on medical histories and ongoing therapies were obtained; blood samples were collected from the antecubital vein for the determination of lipid profile, glucose levels, liver enzymes, leukocyte count and T-cell subsets as previously described[17,38,51,72]. Clinical characteristics are presented in Supplementary Table 1 (supplemental information). Mixed lymphocyte reaction and DC characterization was conducted on a subgroup of 18 subjects, 6 carriers of each apoE2, apoE3 and apoE4 isoforms matched for age and gender, these experiments were performed by a researcher blinded to the apoE genotype.

**Isolation of primary lymphocytes from murine lymphoid organs**. Primary lymphocytes were isolated from the lymph nodes and spleen of apoE KO and WT littermates (the scientist was blinded of the mice genotype). Lymph nodes and spleens removed and weighed. A uniform cell suspension was prepared by washing in 2% FBS, 2 μM EDTA PBS (MACS buffer) and by mashing the lymph nodes and spleens through a 70 μm cell strainer with 1 mL syringe plunger followed by centrifugation at 500×$g$ for 5 min. The supernatant was aspirated and 5 mL (per mice) of red cell blood lysis buffer (Sigma-Aldrich, Cat#R7757) were added for 5 min at 4 °C, to deplete all the red blood cells, washed, spin down at 600×$g$ for 5 min and the supernatant was thrown away. Cells were suspended in 10 mL of MACS buffer and counted.

**Flow cytometry**. All flow cytometry antibodies were used at 1:100 dilutions unless otherwise specified. For immunophenotyping 100 μL of cells suspension and/or blood, or $1 \times 10^6$ were acquired with BD LSR Fortessa (Becton Dickinson) or Novocyte 3000 (ACEA Biosciences). Antibodies used are listed in Supplementary Table 2. The different gating strategies are presented in Supplementary Figure 11-15.

**Purification of CD11c$^+$ DCs and T cells from lymphoid organs**. CD11c$^+$ DCs and T cells were purified from lymph nodes and spleen of C57BL/6 WT and apoE KO mice and BALB/c mice, while CD4$^+$ OTII transgenic T cells and CD8$^+$ OTI transgenic T cells where isolated from the spleen of OTII and OTI TCR-transgenic mice. CD11c EasySep™ Mouse CD11c Positive Selection Kit, EasySep™ Mouse T Cell Isolation Kit, EasySep™ Mouse CD4$^+$ and CD8$^+$ T Cell Isolation Kit (StemCell Technologies, Cat#18758, Cat#19851, Cat#19852 and Cat#19853) were used.

Proliferation was calculated by mitotic index, defined as the ratio between the number of T cells undergoing mitosis to the total number of plated T cells. The index is calculated at each cell division from the sum of mitotic events adjusted for the number of cell precursors.

**Mixed lymphocyte reaction DCs**. Purified T cells were stained with CellTrace™ Violet Cell Proliferation Kit (5 μM, Invitrogen™, Cat#C34557), incubated for 20 minutes at 37 °C in dark, diluted five times in warm media, washed and resuspended in warm media for 10 minutes. $4 \times 10^5$ T cells were cultured with $1 \times 10^5$ of purified DCs and plated with 200 μL of AIM V® Serum Free Medium (Thermo Fisher Scientific, Cat#12055083) in 96-well plate (U-bottom wells) for 5 days at 37 °C with 5% of $CO_2$. Cells were stained with PECF594-conjugated anti-mouse CD4 (BD Biosciences, Cat#562285) and PerCP Cy5.5-conjugated anti-mouse CD8 (eBioscience, Cat#:45-0081-82) for flow cytometric detection of proliferation.

**Antigen-specific T-cell activation**. Purified CD4$^+$ OTII and CD8$^+$ OTI transgenic T cells were stained with CFSE (SIGMA-ALDRICH, Cat#21888), incubated for 10 minutes at RT in dark, diluted ten times in 2% FBS, 2 μM EDTA PBS (MACS buffer), washed three times and resuspended in warm media. $1.5 \times 10^5$ CD4$^+$ OTII or CD8$^+$ OTI transgenic T cells were cultured with $7.5 \times 10^4$ of purified DCs, previously pulsed with OTII or OTI peptide (10 μg/mL, OVA 323–339 and OVA 257–264 [InvivoGen]) for 30 min at 37 °C with 5% of $CO_2$. Cells were then plated with 200 μL of AIM V® Serum Free Medium (Thermo Fisher Scientific, Cat#12055083) in the presence of IL-2 (10 U/mL, Peprotech, Cat#200-02) in 96-well plate (U-bottom wells) for 4 days at 37 °C with 5% of $CO_2$ and proliferation assessed by flow cytometry. For cytokine secretion, after 4 days of co-culture, transgenic OTII CD4$^+$ T cells were pulsed with OTII peptide (1 μg/mL) for 4 h at 37 °C with 5% of $CO_2$ in the presence of Brefeldin A (1:1000, BD Biosciences Cat#554724) and CD28 (5 μg/mL, BD Biosciences Cat#553295). Cytokine staining was performed by flow cytometry following the instructions from the fixation/permeabilization kit (BD Biosciences, Cat#554715), using the Alexa Fluor 647-conjugated anti-mouse IFNγ (BD Biosciences, Cat#563727), PECy7-conjugated anti-mouse IL-4 (BD Biosciences, Cat#560699), BV421-conjugated anti-mouse IL-10 (BD Biosciences, Cat#563276), APC-Cy7-conjugated anti-mouse IL-17A (BD Biosciences, Cat#560881).

**Isolation and culture of bone marrow-derived DCs**. Bone marrow-derived dendritic cells (BMDCs) were generated by flushing with PBS femors and tibia of WT and apoE KO mice or BALB/c mice. Cells were dispersed with a 5 mL syringe and pass through a 70 μm cell strainer, washed with PBS, spin down at 600×$g$ for 5 min and suspended in 5 mL of Red Cell Blood Lysis Buffer (SIGMA-ALDRICH, Cat#R7757) for 5 min at 4 °C to deplete all the red blood cells. Bone marrow cells were washed twice in 2% FBS, 2 μM EDTA PBS (MACS buffer), counted and plated in a six-well plate with AIM V® Serum Free Medium (Thermo Fisher Scientific, Cat#12055083) with 20 ng/mL of GM-CSF (ImmunoTools, Cat#12343125) for 7 days at 37 °C with 5% of $CO_2$. Cells were washed with PBS and refilled with 20 ng/mL GM-CSF in AIM V® Serum Free Medium at day 3 and, at day 6, treated for 18 h with LPS (10 ng/mL, Sigma-Aldrich, Cat#L2630) to induce maturation. For LXR activation, WT and apoE KO matured BMDCs were treated with the LXR agonist GW3965 (1 μM) overnight at 37 °C with 5% of $CO_2$.

**EαGFP assay**. In vitro assay was performed with immature BMDCs at day 6 treated with EαGFP peptide (50 μg/mL) for 4 h at 37 °C with 5% of $CO_2$. BMDCs were then collected, washed twice with 2% FBS, 2 μM EDTA PBS (MACS buffer) and stained with biotin-conjugated anti-mouse Eα 52–68 peptide (eBioscience, Cat#13-5741-81), for 30 min at 4 °C. Cells were washed twice with 2% FBS, 2 μM EDTA PBS (MACS buffer) and stained with APC-conjugated streptavidin (eBioscience, Cat#17-4317-82) for 30 min at 4 °C. Antigen uptake and presentation were assessed by flow cytometry.

For in vivo assay, WT and apoE KO mice were injected i.p. with EαGFP peptide (20 μg/mL). After 24 h, peritoneal lavage was collected, washed in 2% FBS, 2 μM EDTA PBS (MACS buffer) and stained for flow cytometry analysis with eFluor660-CD11c (eBioscience, Cat#500114) and eFluor 450- MHCII (eBioscience, Cat#48-5321-82).

**Cytokine production**. Cytokine production was assessed in BMDCs (day 6) treated with LPS (10 ng/mL SIGMA-ALDRICH, Cat#L2630) for 4 h in the presence of Brefeldin A (1:1000, BD Cat#554724). Cytokine staining was performed by flow cytometry following the instructions from the fixation/permeabilization kit (BD, Cat#554715), using the Alexa Fluor 647-conjugated anti-mouse IFNγ (BD, Cat#563727), PECy7-conjugated anti-mouse IL-4 (BD, Cat#560699), BV421-conjugated anti-mouse IL-10 (BD, Cat#563276), Alexa Fluor 647-conjugated anti-mouse IL-23 (BD, Cat#565317), FITC-conjugated anti-mouse IL-12 (BD, Cat#560564).

**Bone marrow transplantation**. WT and apoE KO mice were lethally irradiated with a total dose of 900 cGy. Two hours later, mice were injected in the tail vein with $5 \times 10^6$ nucleated bone marrow (BM) cells from WT or apoE KO donors.

Recipient mice received gentamycin (0.4 mg/ml in drinking water) starting 10 days before irradiation and maintained thereafter. At 10 weeks after BM transplantation, $K^d$ skin allograft transplantation was performed as described below.

**Skin graft allotransplantation**. Skin grafting was conducted using a method previously described by Billingham and Medawar, 1951 using tail skin grafted onto the lateral thorax. Donor tail skin was removed and cut into 1 $cm^2$ sections. When recipient mice reached a surgical plane of isoflurane anaesthesia, 0.26 µg of Metacam (Boehringer Ingelheim GmbH) was injected s.c. Bandages were removed 7 days post transplantation and mice were observed for rejection for 3 weeks by an observer blinded to the mice genotype.

**$K^d$ skin allograft**. C57BL/6 WT and apoE KO littermates and bone marrow transplanted WT and apoE KO mice (both males and females) received skin grafts from female C57BL/6$K^d$ mice.

**Male skin allograft**. C57BL/6 WT female received a C57BL/6 WT or apoE KO male-derived skin graft.

**Migration assay**. Transwell 5 µm pore membranes (Corning®, Cat#CLS3421) were used in the assay putting in the top chamber, a cell suspension of 300 µL of $0.6 \times 10^6$ lymphocytes, isolated from draining lymph nodes of grafted mice, while in the bottom chamber, a solution of 700 µL of R10 medium plus CXCL10 or CCL19/CCL21 (200 ng/mL, Peprotech, Cat#250-27B and Cat#250-13). Cells were incubated at 37 °C with 5% of $CO_2$ and counted manually, with a counting chamber, after 2, 4, 6 h.

**Cholesterol determination and FPLC**. Blood samples were collected in EDTA tubes by intracardiac puncture and plasma was separated by high speed centrifugation at 4 °C. For FPLC analysis, 200 µL pooled plasma ($n = 4$ mice total) was separated on a Superose column (Amersham) at a flow rate of 0.4 mL/min. Fractions were collected and analyzed for total cholesterol quantification by standard enzymatic techniques using the Cholesterol CP Kit (ABX Pentra, Cat#A11A01634) (Sala et al., 2014).

**Lipidomic profiling of T cells and DCs**. Lipidomic profiling of T cells and DCs focused on phospholipids, fatty acids and sterols and was performed as described in Cermenati et al. 2015. Out of 356 phospholipids were analyzed, including LysoPC: lysophosphatidylcholine; PC: phosphatidylcholine; PC aa: phosphatidylcholine acyl-acyl; PC ae: phosphatidylcholine acyl-alkyl also known as plasmalogens; PE: phosphatidylethanolamine; PE aa: phosphatidylethanolamine acyl-acyl; PE ae: phosphatidylethanolamine acyl-alkyl also known as plasmalogens; PS: phosphatidylserines; LysoPI: lysophosphatidylinositol; PI: phosphatidylinositol; PG: phosphatidylglycerol; LysoPA: lysophosphatidic acid; PA: phosphatidic acid; SM: sphingomyelin; Cer: ceramide; GCer: glucosyl/galactosyl-ceramide; LacCer: lactosylceramide and gangliosides GM1, GM2 and GM3) 229 different species belonging to different families were detected. Fifteen fatty acids were detected: palmitic acid C16:0; palmitoleic acid C16:1; margaric acid C17:0; stearic acid C18:0; oleic acid C18:1; linoleic acid C18:2; γ-linolenic acid C18:3; arachic acid C20:0; arachidonic acid C20:4; eicosapentaenoic acid C20:5 (EPA); behenic acid C22:0; erucic acid C22:1; docosahexaenoic acid C22:6 (DHA); lignoceric acid C24:0; nervonic acid C24:1. Cholesterol, desmosterol and seven oxysterols (24-hydroxycholesterol, 25-hydroxycholesterol, 27-hydroxycholesterol, 24,25-epoxycholesterol; 7α-hydroxycholesterol, 7β-hydroxycholesterol, 7-Keto-cholesterol) were detected.

**Reagents**. All HPLC solvents were mass spectrometry grade (Carlo Erba Reagents, Italia). The internal standard for cholesterol and oxysterols quantification was cholesterol-2,2,3,4,4,6-$d_6$ (SIGMA-ALDRICH) for fatty acids internal standards were heneicosanoic acid (C21:0) and uniformly labeled 13C-linoleic acid (C18:2) (SIGMA-ALDRICH). Phospholipid standards: C13:0 lysophosphatidylcholines (LysoPC); C25:0 phosphatidylcholines (PC); C12:0 sphingomyelin (SM); 12:0–13:0 phosphatidylserine (PS); 12:0–13:0 phosphatidylinositol (PI); 12:0–13:0 phosphatidylglycerol (PG); 12:0–13:0 phosphatidic acid (PA); 12:0–13:0 phosphatidylethanolamine (PE); C12 ceramide (Cer); glucosyl (β) C12 ceramide (GC); lactosyl (β) C12 ceramide (LacCer); C17 mono-sulfo galactosyl(β) ceramide (D18:1/17:0; GalCer) were purchased from Avanti Polar Lipids.

**Phospholipid, FA, and cholesterol and oxysterol extraction methods**. Internal standards were added to samples, and lipid extraction was performed using the Folch method [chloroform–methanol (MeOH) 2:1, v/v]. Samples were left overnight at 4 °C. Then the organic residue was divided in two fractions: one for the analysis of phospholipids (fraction A, 60% of the total sample), and the other for analysis of total FAs, cholesterol and oxysterols (fraction B, 40% of the total sample). Total FAs, cholesterol and oxysterols were obtained from samples by acid hydrolysis. Fraction B was resuspended in chloroform–MeOH 1:1, v/v. 1 M HCl:MeOH (1:1, v/v) was added to the total lipid extract and shaken for 2 h. Then chloroform-water (1:1, v/v) was added, and the lower organic phase was collected, split, transferred into tubes and

dried under nitrogen flow. The residue was resuspended in 1 mL of MeOH and split 60/40 for total FA and total cholesterol analysis, respectively.

**Fatty acid and phospholipid profile**. For total fatty acid quantification, aliquots of each sample (10 µL) were diluted 1:10 in MeOH/water (50:50 v/v), transferred into a 96-well-plate and placed in an auto-sampler for LC–MS/MS analysis. Quantitative analysis was performed with calibration curves with pure standards prepared and analyzed daily by electrospray ionization (ESI) using an API 4000 triple quadrupole instrument (AB Sciex, USA). The LC mobile phases were: water/10 mM isopropylethylamine/15 mM acetic acid (phase A) and MeOH (phase B). The gradient (flow rate 0.5 ml/min) was as follows: T 0: 20% A, T 20: 1% A, T 25: 1% A, T 25.1: 20% A, T 30: 20% A. The Hypersil GOLD C8 column (100 mm × 3 mm, 3 µm) was maintained at 40 °C. The mass spectrometer was operated in selective ion monitoring (SIM)/SIM mode. Data processing was performed using MultiQuant software (AB Sciex, USA).

For the quantification of the different phospholipids the MS analysis was performed with a flow injection analysis-tandem mass spectrometry (FIA-MS/MS) method. The identity of the different phospholipid families was confirmed using pure standards, namely one for each family. Methanolic extracts were analyzed by a 3 min run in both positive and negative ion mode with a 268 multiple reaction monitoring (MRM) transition in positive mode and 88 MRM transition in negative mode. An ESI source connected with an API 4000 triple quadrupole instrument (AB Sciex, USA) was used. The mobile phase was 0.1% formic acid in MeOH for FIA positive analysis and 5 mM ammonium acetate pH 7 in MeOH for FIA negative. MultiQuant™ software version 3.0.2 was used for data analysis and peak review of chromatograms. Semiquantitative evaluation of phospholipids was performed using MultiQuant software based on external pure standards.

**Cholesterol and oxysterols quantification**. Fractions for the quantitative analysis of total cholesterol and oxysterols, were analyzed by positive atmospheric pressure chemical ionization (APCI+) using a linear ion trap-mass spectrometer (LTQ, Thermo Fisher Co., San Jose, CA, USA) using nitrogen as sheath, auxiliary and sweep gas. The instrument was equipped with a Surveyor liquid chromatography (LC) Pump Plus and a Surveyor Autosampler Plus (Thermo Fisher Co., San Jose, CA, USA). The mass spectrometer was employed in MS/MS mode using helium as collision gas. The LC eluents were (phase A) acetonitrile (ACN)/methanol (MeOH)/water ($H_2O$) (76;20;4, v/v) and (phase B) propan-2-ol (IPA). The gradient (flow rate 1 ml/min) was as follows: T0 100% A, T15 100% A, T15.50% A, T35 50% A, T35.50 100% A, T50 100% A. The Inertsil ODS-2 150 mm × 4.6 mm, 5 µm (GL Sciences, Tokyo, Japan) column was maintained at 30 °C. Quantitative analysis was performed on the basis of calibration curves with pure standards.

**Quantitative real time PCR**. NucleoSpin® RNA kit (Macherey-Nagel, Cat#740955.50) was used to perform the RNA extraction from purified dendritic cells, T cells, BMDCs, peritoneal macrophages and liver from WT and apoE KO mice according to manufacturer's instructions. The kit was also used for mRNA extraction from human DCs of carriers of different apoE isoforms. RNA assessed for quality and quantity using absorption measurements (NanoDrop™ 1000 Spectrophotometer, Thermo Fisher Scientific) and transcripted in cDNA with iScriptTM cDNA synthesis kit (BioRad). Gene expression analysis was performed using SYBR Green Supermix (Thermo Fisher Scientific) in CFX connect light cycler (BioRad, Cat#1708841). Expression was calculated using the ΔΔCt method (Livak and Schmittgen, 2001) and normalized to a housekeeping gene (RPL for mouse genes and β-actin for human genes). Primers for qPCR were designed with the help of online tools (https://www.eurofinsgenomics.eu/). The thermal cycling profile was a two-step amplification (95 °C for 5 min, followed by 45 cycles of 95 °C for 10 s and 55 °C for 30 s). The sequences of the qPCR Primers are listed in Supplementary Table 3.

**Fluorescence and confocal microscopy**. Cells were allowed to adhere onto poly-L-lysine coated and fixed in 4% formaldehyde PFA (SIGMA-ALDRICH, Cat#F8775) for 15 min at room temperature. Cells were then washed in PBS, blocked in blocking buffer (PBS containing 5% BSA, SIGMA-ALDRICH) and then stained with filipin (0.05 mg/mL SIGMA-ALDRICH, Cat#F9765) at 37 °C for 1 h, CTXb (Cholera Toxin B subunit FITC conjugate—8 µg/mL, SIGMA-ALDRICH, Cat#C1655) or appropriate primary antibodies (APC-conjugated anti-MHC Class II I-A/I-E, eBioscience Cat#48-5321-82) at 4 °C for 30 min in the dark. Following staining, cells were washed with PBS and coverslips were mounted onto slides using Fluoroshield™ with DAPI (SIGMA-ALDRICH, Cat# F6182) and then examined using a Zeiss microscope equipped with ×40 objective or Leica SP5 confocal microscope equipped with a 63 × 1.4 NA objective.

Confocal images were acquired and analysed by Leica LAS software. Analysis of fluorescence intensity and co-localization was performed with ImageJ.

**DC-T-cell conjugate analysis**. Freshly isolated or 5 day-activated (with plastic-bound anti-CD3/28 [eBioscience, Cat#16-0032-86 and Cat#16-0281-81] plus 50 U/mL IL-2 [Roche, Cat#16-0281-81]) lymphocytes from lymph nodes of WT and apoE KO mice were stained with 1 µM CFSE (Sigma-Aldrich, Cat#21888), washed three times in 2% FBS, 2 µM EDTA PBS (MACS buffer) and resuspended $10^6$/mL. Dendritic cells were cultured from bone marrow of BALB/c mice, collected after

6 days, stained with 0.5 μM DDAO (Invitrogen, Cat#C34553) washed three times in 2% FBS, 2 μM EDTA PBS (MACS buffer). 100 μL of lymphocyte suspension ($10^5$ cells) were incubated with DCs at different ratio (1:3 and 1:5 DCs:T cells) for 15 min at 37 °C, gently resuspended and fixed for flow cytometry analysis.

**pAKT phosphorilation**. Purified T cells were plated $0.5 \times 10^6$ per well and allowed to settle, then 1 μg/mL biotin labeled anti-CD3 and anti-CD28 (Thermo Fisher Scientific, Cat#11456D) added for 5 min at 4 °C and crosslinked with strepatvidin (20 μg/mL). At 1,5 and 15 min time points 4% PFA was added to cell mixture and fixed at 27 °C for 15 min. Cells were washed with PBS, permeabilized with 100% ice-cold methanol and placed at −20 °C for 10 min. Permeabilized cells were incubated in blocking buffer (2% FCS/PBS + $Na_3VO_4$ + NaF + NaCl) at RT for 30 min. Cells were then stained with BV605-conjugated anti-mouse CD4 (Biolegend, Cat#100547), PE-conjugated anti-mouse CD8 (eBioscience, Cat#12-0081-82) and 1:150 diluted anti-phospho-Akt (S473) (Cell Signaling Technology, Cat#9271) antibody for 30 min at RT. Cells were washed twice with PBS and stained with 1:4000 Alexa Fluor 647 Goat anti-rabbit IgG (Life Technologies, Cat# A21245) at RT for 30 min. Cells were washed twice with PBS and resuspended in 200 μL PBS for cytofluorimetric analysis.

**Isolation and culture of MDCs and T cells from human blood**. For human study, 15 mL of blood (supplemented with EDTA) were centrifuged for 12 min at 1000×g. Plasma was discarded and the interface between plasma and red blood cells, enriched in leukocytes and platelets (buffy coat), was carefully collected, diluted with cold PBS and stratified on 3 mL of Ficoll-Plaque™ PREMIUM (GE-Healthcare, Cat#17-5442-03). After centrifugation of 35 min at 250×g, PBMC layer was carefully collected and washed three times with 10 mL of cold PBS at 180×g for 12 min, to get rid of platelets. For MDCs growing, PBMC were plated in six-well plate with AIM V® Serum Free Medium for 2 h at 37 °C and 5% $CO_2$ to allow adhesion of monocytes. Later, cells were washed twice with PBS at RT, and suspended in AIM V® Serum Free Medium supplemented with 5% of autologous/allogenic serum and GM-CSF (200 U/mL) and IL-4 (300 U/mL). After 5 days, LPS (1 μg/mL, Sigma-Aldrich, Cat#L2630) was added and cells collected 48 h later. Cells were washed in 2% FBS, 2 μM EDTA PBS (MACS buffer), and then stained with filipin (0.05 mg/mL Sigma-Aldrich, Cat#F9765) at 37 °C for 1 h, CTXb (Cholera Toxin B subunit FITC conjugate—8 μg/mL, Sigma-Aldrich, Cat#C1655) or FITC-conjugated anti-human HLA-DR (BD, Cat#347363) or CD80 (BD, Cat#560926) 4 °C for 30 min in the dark. Following staining, cells were washed with 2% FBS, 2 μM EDTA PBS (MACS buffer) and analysed by flow cytometry.

Naive $CD4^+$ T cells were isolated from PBMC with Naive CD4+ T Cell Isolation Kit II, human (Miltenyi Biotec, Cat#130-094-131).

**Co-culture of T cells and MDCs**. For human mixed lymphocyte reaction, $CD4^+$ T naive cells were co-cultured with autologous or allogenic monocyte-derived dendritic cells in AIM V® Serum Free Medium supplemented with 5% of allogenic serum and GM-CSF (200 U/mL) and IL-4 (300 U/mL) at 37 °C with 5% of $CO_2$. After 3 or 5 days, cells were collected and stained for FITC-conjugated anti-human CD45RA (BD Biosciences, Cat#555488) and PE-conjugated anti-human CCR7 (BD Biosciences, Cat#552176).

**Statistical analysis**. Graph Pad-Prism6 was used for graphic presentation and statistical analysis. Results are given as the mean per group ± SEM. The data were analysed according to what reported in figure legends and a *p*-value of <0.05 was considered significant. For comparison between two groups unpaired two-sides *T*-test with a 95% confidence interval was used; for multiple comparisons a regular two-way ANOVA with a 95% confidence interval was used.

The qPCR data were analyzed using the delta CT method by taking the CT values of the genes of interest from the housekeeping gene following by normalization to the wild type control sample.

**Data availability**. The authors declare that the data supporting the findings of this study are available within the article and its supplementary information files, or are available upon reasonable requests to the authors. Further information and requests for reagents may be directed to, and will be fulfilled by the corresponding author Giuseppe Danilo Norata or by Fabrizia Bonacina (danilo.norata@unimi.it; fabrizia.bonacina@unimi.it).

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

## Acknowledgements
EaGFP plasmid was a gift from Prof. Jenkins, University of Minnesota. We acknowledge Luca Cesana for his help for the experiments with OT transgenic mice. The work of the authors is supported by: Fondazione Cariplo 2015-0524 and 2015-0564 (A.L.C.) and 2016-0852 (G.D.N.); H2020 REPROGRAM PHC-03-2015/667837-2 (A.L.C.); Telethon Foundation (GGP13002) (G.D.N.), Ministero della Salute GR-2011-02346974 (G.D.N.) and GR-2013-02355011 (F.B.); Aspire Cardiovascular Grant 2016-WI218287 (G.D.N.). F.M.M.-B. is a recipient of the British Heart Foundation Chair of Cardiovascular Immunology (CH/15/2/32064).

## Author contributions
Conceived and designed the experiments: F.B., D.Coe., A.L.C., F.M.M.-B. and G.D.N. Performed the experiments: F.B., D.Coe., G.W., A.B., A.M., K.G., P.U., F.P., L.G. and L.D. Analyzed the data: F.B., D.Coe., D.Caruso., N.M., F.M.M.-B. and G.D.N. Contributed reagents/materials: M.P.L., Q.X., A.A. and S.G. Wrote the paper: F.B., F.M.M.-B. and G.D.N.

## Additional information

**Competing interests:** The authors declare no competing interests.

