## [Peer Review File · Nature Communications]

Reviewers' comments:

Reviewer #1 (Remarks to the Author):

In the study from Bonacina and colleagues the role of ApoE in DC biology is studied. They demonstrate that cell intrinsic ApoE expression by DCs shapes their CD4 priming ability. Mechanistically, their data suggest that ApoE through regulation of cholesterol homeostasis controls lipid raft formation and thereby antigen presentation. The presented data, obtained from an elegant set of murine models and as well DCs from humans with different ApoE variants, are convincing and largely support the overall conclusions. Moreover, a direct intrinsic role for ApoE in affecting DC function has not been described before and shed new light on the well-known association between hyperlipidaemia and immune activation. Therefore the study is novel and potentially interesting to the readership of Nature communications. However I have some comments that need to be addressed and issues that need to be clarified to strengthen the paper.

Major comments:

- 1) Based on Figure 2B the authors conclude that deficiency in ApoE in haematopoietic compartment rather than somatic deficiency can explain the rejection phenotype. However, KO BM -> KO mouse (black squares) seems to give a more rapid rejection than KO BM -> WT mouse (gray circles). Wouldn't that suggest that ApoE deficiency outside of the haematopoietic compartment also contribute to this phenotype? The authors should comment on this.
- 2) In Figure 2E-F the authors use BM chimeras (KO BM -> WT mice versus WT BM -> KO mice) to test the hypothesis that 'allogenic skin rejection is independent of the systemic metabolic environment and the consequence of a specific effect of leukocyte ApoE deficiency.' However the comparison of these two chimeras is not logical in this context. The right approach would have been to test WT or KO BM on the same genetic background of the recipient (eg WT BM -> WT mice versus KO BM -> WT mice or WT BM -> KO mice versus KO BM -> KO mice). That is a only way to really test whether leukocyte ApoE deficiency is important irrespective of metabolic environment.
- 3) In figure 3 the authors use a HLA mismatch approach to assess T cell priming differences between WT and KO DCs. This is an important dataset that needs to be further validated using a antigen specific (non mismatch-based) readout. For instance by using OVA pulsed DCs and coculture with OTI and OTII cells.
- 4) In figure 4E and suppl Fig 6 the authors analyse activation marker expression of splenic WT and KO DC subsets. What is expression of MHCII on cDC2 and what is expression of MHCII and MHCII on cDC1? Does that correlate with the selective change in CD4 priming? If only an effect of MHCII on Th cell-priming cDC2 is found, and no change on cDC1s that are better at priming CD8 responses, than that would strengthen the overall message of a selective effect on CD4 priming ability in the absence of ApoE.
- 5) The authors hypothesize that increased lipid raft formation in ApoE KO DCs allow for more surface MHCII expression. Is this redistribution intracellular pools of MHCII to the surface or also a result of more overall MHCII protein expression. To address this, the authors should perform a simple but insightful analysis by doing a surface staining MHCII or HLADR versus a intracellular staining of those markers in WT and KO DCs.
- 6) The authors convincingly show that CD4 priming is enhanced due to loss of ApoE or expression of ApoE4. However, it is unclear whether the Th cell polarization is skewed in one particular direction which may additionally explain increased graft rejection for instance. Therefore the authors should analyse IFNg, IL17 IL4 and IL10 expression by the T cells to assess T cell polarization. Along the same vein, the authors analysed DC surface marker expression. What about DC cytokine release? Such as IL12, IL10, IL6, IL23?
- 7) Figure 6 and 7 display very interesting data. However, it would be further strengthened if in figure 6K the reverse experiment would also be performed: culture ApoE2 DCs with T cells from ApoE2, -3 and -4 donors. Of there is no phenotype there, it would strengthen the claim that also in humans the ApoE effect on immune cell activation is driven by changes selectively in DCs.

Minor comments:

- 1) Line 171-174: The authors use the terms conventional versus lymphoid DCs to refer to CD11b+ and CD8+ splenic DCs. However, that is an outdated nomenclature. Better use conventional (c)DC1 and cDC2, respectively.
- 2) Although the approach to come to the conclusion that the phenotype can be largely explained change in DC biology is valid, it is somewhat indirect. Could the authors comment on why they have not used CD11c-Cre / ApoE flox mice?
- 3) The title of Fig 2 is incorrect. There is no data in this figure that point towards a role of myeloid deficiency of ApoE in promoting rejection. Only that it is haematopoietic.
- 4) Using the word 'Defines' in the title seems a little too strong – as if that is the only factor. I would suggest to change that to 'regulates' or 'control'

Reviewer #2 (Remarks to the Author):

In experiments spanning the in vitro-mouse-human spectrum Bonacina et al study the immunoregulatory role of ApoE in DCs. This is an elegant study that expands our understanding of ApoE's role in innate and adaptive immunity. Furthermore, the role of cholesterol homeostasis in DCs has garnered attention in the last couple of years, and as such this study is timely. Overall, the experiments are well designed and controlled, and the results largely support the main conclusion that ApoE deficiency in DCs alters their immune-stimulatory function.

Major:

1) The consequence of ApoE-deficiency on cellular sterol homeostasis is somewhat perplexing. The increased level of total cellular cholesterol and of several species of oxysterols (fig 5 and S7,S8) is not reflected in SREBP- and/or the LXR-regulated gene program (Fig 5c; Note: cd36 does not respond to cholesterol and the comment alluding to this in the text should be corrected). Increased accumulation of these lipids is expected to significantly decrease SREBP targets (HMGCR is not reduced) and increase LXR targets (ABCA1 and G1 are not increased). The authors invoke the argument that desmosterol is reduced, however, as shown in fig 5A and S7 this is not significant so this cannot explain the lack of change in gene expression. The authors should therefore evaluate a larger set of target genes and also consider determining the protein levels of a select number of these targets in order to draw a clearer conclusion on this. In line with this point, does pharmacological LXR activation reverse lipid accumulation, inflammatory signaling and hyperactivity of ApoE null DCs? This seems relevant since it will address the role of cholesterol accumulation and of ApoE - an LXR target - in the DC phenotype.

2) The analysis of lipid rafts in ApoE-KO DCs is limited. Given that in the study it is reported that ApoE KO DCs have higher MHC-II and lipid rafts (as assessed by CTXb staining) it is not surprising that when using a threshold in their analysis one would find an increased co-localization coefficient. Addressing causality may be difficult, but at the very least the authors need to evaluate whether the effect is MHC-II-specific, or that other lipid-raft associated signaling events are altered in the ApoE KO cells. For example, are surface TLRs or RTKs and their associated signaling also changed?

3) The attempt to translate the mouse-based findings to the human setting is commendable yet the analysis of the human DCs is rudimentary. That ApoE influences Cd4 T-cell polarization in the MLR assay (Fig 6K) is convincing. However, the evidence that this is related to the sterol-status of these cells, or to ApoE-dependent lipid transport is lacking. Are cellular sterol levels affected in the DCs in an ApoE-isoform dependent manner? Is expression of cholesterol-related genes altered? Is bona fide cholesterol efflux different in these DCs?

Minor:

- The article will benefit from English proof reading as well as paying attention to punctuation and use of consistent tense (also in methods and sup. Material sections). This will make readability easier. Here and there one comes across typos. For example: in Figure 1F “deplition” > depletion. Please check.
- The figure legends are concise to the point that it’s not easy to follow some of the experiments. Some experimental information would be helpful. Also, the legends should clearly state the group/sample size as this is not always indicated.
- The statistics section in the Materials and Methods indicates that paired analysis was used, while the legends report unpaired. Please check and clarify.
- It would be informative if the authors comments on the Cd4/8 differential effect of ApoE in DCs.
- In the graft experiments, a cut-off of 50% skin-surface rejection is considered as a full rejection. Please clarify the rational for using this (arbitrary) cut-off value.
- Figure 5: scale bars in IF images are missing.
- The in vivo EaGFP procedure is missing in the methods section. Please add.
- The paragraph discussing LXR in the Discussion section is unclear and confusing (lines 283-98) and should be clarified. Furthermore, the argument invoking the role of ABCA1 as an anti-inflammatory determinant of LXR activation should take into account results from the Tall group that find that LXR activation is anti-inflammatory also in Abca1/G1-null cells.
- Figure S8A,B is not clear. What is being compared and between which cells?
- L.130: The statement made in this line seems wrong since hypercholesterolemia is largely corrected in the KO mice receiving WT marrow. Please correct or clarify.
- The change in color coding of the legend in fig 7 is confusing.

Reviewer #3 (Remarks to the Author):

This study investigates the role of ApoE in dendritic cell function and adaptive immune responses. It builds on previously published findings showing increased CD4+ Teff cells in ApoE ko mice along with a reported role for ApoA1 in MHC II presentation and T cell activation (ref 9). A direct role of ApoE on DC function in mouse and humans and the mechanisms surrounding it would perhaps not be unexpected, however would be of interest and relevance to the field. The study uses ApoE ko mice as well as human blood donors expressing different ApoE isoforms, which are both great models and valid approaches. However, a more comprehensive array of DC functional assays is required to provide a convincing role for ApoE on DC function and the mechanisms by which it acts to enhance CD4+ T cell numbers, particularly antigen processing and presentation.

1. The conclusion that allograft rejection is independent of a systemic inflammatory response (line 126) is based on cholesterol levels (Fig 2C), which do not appear to correlate with allograft rejection (Fig 2B). This is indirect and evidence of similar levels of inflammatory mediators would be required to demonstrate independence of systemic inflammatory responses in allograft rejection, especially since irradiation and the allo reaction itself are also likely contributing to systemic inflammatory responses in these models.

2. Fig 2E shows that ApoE deficiency within the BM compartment rather than the T cells themselves are important. The Fig 2 title should state “BM deficiency of ApoE” since a direct role for myeloid cells has not been shown. Likewise “an immunomodulatory role for myeloid but not hepatic apoE” (abstract lines 28-29) is not well supported by the data. Cells other than myeloid cells are derived from transferred BM and myeloid cells in the liver may also be contributing.

3. ApoE ko mice more readily reject allografts (Fig 1F) and conversely, ApoE ko grafts are more readily rejected (Fig 3J), suggesting ApoE deficiency in both donor and host are important in allograft rejection. However, these data do not demonstrate a direct role for DC in graft survival and hence the conclusions that “lack of apoE expression by graft-derived DC was responsible” (lines 162,163) is not supported by the data. Demonstration of a role for DC in this model would require a strategy (eg depletion Abs, DTR mouse models) to remove DC.

4. Fig 3 C-G: What is CD4 mitotic index?? How was this calculated??

5. ApoE ko mice have increased numbers of DC and increased MHC II expression. However they

also have significantly lower levels of activation marker CD80 (Fig S6) which is at odds with the statement that ApoE deficiency is not associated with costimulatory function (line 178). In order to make this conclusion expression of costimulatory molecules and DC cytokine secretion should be examined in the presence or absence of TLR ligands or other well-defined DC stimuli.

6. Details of the ag uptake and processing assay are missing from the methods and the dot plots in Fig 4 are not convincing. The percentage of GFP+Ea:IAb+ events for both WT and ApoE KO are <2% in the representative dot plots in Fig 4G yet over 10% in all of the data points in Fig 4F.

Relevant control plots should be shown. The inclusion of other assays such as DQ-OVA fluorescence and or and processing and presentation of Ag to Ag-specific T cells would provide more convincing evidence for a direct role of ApoE in antigen processing and presentation.

7. Fig 5C what is the fold of expression relative to?

8. The conclusions that increased CD4+ TEM polarization in humans carrying the apoE4 isoform is due to enhanced antigen presentation activity by DC (line 245) is not well supported by the experimental design shown in Fig 6J. In this experiment, autologous MDC and naïve CD4+ T cells are co-cultured in the absence of exogenous antigen. Under these conditions T cells would not be expected to proliferate at all and both T cells and DC would be expressing the same ApoE4 isoform. It therefore isn't clear whether the T cell phenotype changed during the culture and therefore whether autologous DC in the steady state had any influence on this. To test whether antigen uptake and processing is altered by the different isoforms expressed by MDC DQ-OVA or similar functional assays would need to be performed. For CD4 T cell proliferation and differentiation of effector function, MDC from donors expressing different ApoE isoforms would need to be co-cultured with allogeneic CD4 T cells, preferably from a donor expressing the parent ApoE3 isoform. Performing these experiments in the presence and absence of the apoE3-HDL would further increase the quality of the data.

9. The data in Fig 7 using human MDC shows % of DC expressing HLA-DR and CD80. All DC should express HLA-DR. Representative stains, gating strategies and isotypes or FMO controls should be shown in the supplementary data. The data in the mouse models shows changes in the intensity of MHC Class II and CD80 expression rather than percentage of expressing cells. A similar analysis of the human data could be informative.

10. The published literature describing ApoE and the phenotype of the ApoE ko mice in discussion (lines 259-269) should be moved to the introduction.

Reply to comments from Reviewer #1

Dear Reviewer #1, in this revised version of the paper, we addressed all comments raised by you and by the other reviewers. More in details we have 1) performed antigen presentation experiments by pulsing DCs with OT-I or OT-II peptides and tested in vitro proliferation of OTI and OTII cells from the two transgenic animal models to specifically address MHC-I or MHC-II dependent antigen presentation; 2) improved the characterization of activated DCs in apoE KO mice and investigated the cytokine signature and expanded the analysis of the sterols signature by performing a more comprehensive analysis of genes and proteins related to cholesterol/oxysterol metabolism; 3) tested the distribution of intracellular vs membrane MHC-II and investigated the expression of other lipid raft-associated receptors, such as TLR4; 4) expanded the metabolic characterization of human MDCs and assessed the expression of genes related to cholesterol/oxysterol metabolism; 5) investigated functional differences in DC and T cells from carriers of the apoE2, apoE3 and apoE4 isoforms.

Regarding your specific comments, we offer the following replies:

Major comments:

1) Based on Figure 2B the authors conclude that deficiency in ApoE in haematopoietic compartment rather than somatic deficiency can explain the rejection phenotype. However, KO BM -> KO mouse (black squares) seems to give a more rapid rejection than KO BM -> WT mouse (gray circles). Wouldn't that suggest that ApoE deficiency outside of the haematopoietic compartment also contribute to this phenotype? The authors should comment on this.

Thank you for bringing up this point. Although the rejection phenotype in KO BM -> KO mouse is not significantly different from that observed KO BM -> WT mouse, we cannot exclude the possibility that the very high levels of plasma cholesterol observed in KO BM -> KO favor systemic inflammation with in turn might sustain a more rapid rejection. This possibility is now discussed in the text, page 6 last paragraph. However, the significantly faster rejection on KO BM -> WT mouse vs WT BM -> KO mouse clearly indicates a non-redundant role of ApoE in the hematopoietic compartment.

2) In Figure 2E-F the authors use BM chimeras (KO BM -> WT mice versus WT BM -> KO mice) to test the hypothesis that 'allogenic skin rejection is independent of the systemic metabolic environment and the consequence of a specific effect of leukocyte ApoE deficiency.' However the comparison of these two chimeras is not logical in this context. The right approach would have been to test WT or KO BM on the same genetic background of the recipient (eg WT BM -> WT mice versus KO BM -> WT mice or WT BM -> KO mice versus KO BM -> KO mice). That is a only way to really test whether leukocyte ApoE deficiency is important irrespective of metabolic environment.

Thank you for your suggestion, we now present data also for WT BM -> WT mice (Figure 2E to 2H, see below) to show that leukocyte ApoE deficiency is important irrespective of the genetic background of the recipient. This is now discussed in the text, page 7 first paragraph.

3) In figure 3 the authors use a HLA mismatch approach to assess T cell priming differences between WT and KO DCs. This is an important dataset that needs to be further validated using an antigen specific (non mismatch-based) readout. For instance by using OVA pulsed DCs and coculture with OTI and OTII cells.

To address this suggestion, we performed an antigen-specific assay by coculturing DC from WT or apoE KO mice, pulsed with OT-I or OT-II cognate peptide with either CD8⁺ T cells from OT-I transgenic mice or CD4⁺ T cells from OT-II transgenic mice. The data show that cell proliferation was significantly different when OT-II T cells were co-cultured with apoE deficient DCs compared to WT, while the rate of proliferation was similar when OT-I T cells were co-cultured with apoE KO or WT DCs. These data are now discussed in the text, page 8, first paragraph and, are shown in Figure 3F to 3H and below.

4) In figure 4E and suppl Fig 6 the authors analyse activation marker expression of splenic WT and KO DC subsets. What is expression of MHC I on cDC2 and what is expression of MHCII and MHC I on cDC1? Does that correlate with the selective change in CD4 priming? If only an effect of MHCII on Th cell-priming cDC2 is found, and no change on cDC1s that are better at priming CD8 responses, than that would strengthen the overall message of a selective effect on CD4 priming ability in the absence of ApoE.

In response to this comment we have investigated MHC-I and MHC-II expression in both cDC1 and cDC2 cells. This analysis confirmed that splenic cDC1 from WT and apoE KO mice express similar levels of MHC-I and MHC-II while cDC2 from apoE KO mice present an increased expression of MHC-II but not MHC-I compared to cDC2 from WT mice. These data are now discussed in the text, page 9 line first paragraph and, presented in figure 4D to 4E and below.

5) The authors hypothesize that increased lipid raft formation in ApoE KO DCs allow for more surface MHCII expression. Is this redistribution intracellular pools of MHCII to the surface or also a result of more overall MHCII protein expression. To address this, the authors should perform a simple but insightful analysis by doing a surface staining MHCII or HLADR versus a intracellular staining of those markers in WT and KO DCs.

Thank you for your suggestion, we have investigated surface and intracellular MHCII expression in cDC2 cells and observed that only surface expression of MHC-II was increased in cDC2 cells from apoE KO mice compared to WT while intracellular expression was similar (see below).

6) The authors convincingly show that CD4 priming is enhanced due to loss of ApoE or expression of ApoE4. However, it is unclear whether the Th cell polarization is skewed in one particular direction which may additionally explain increased graft rejection for instance. Therefore the authors should analyse IFN γ , IL17, IL4 and IL10 expression by the T cells to assess T cell polarization. Along the same vein, the authors analysed DC surface marker expression. What about DC cytokine release? Such as IL12, IL10, IL6, IL23?

This is an important point. We have analyzed the expression of IL-4, IL-10, IL-17 and IFN γ in OT-II T cells incubated with WT or ApoE KO dendritic cells pulsed with OT-II peptide and confirmed an increased IFN γ production in both antigen-specific and mismatch-based MLR assay. These data are now presented in Supplemental Figure S5A-S5B and below.

As per your suggestion, we have also investigated IL12, IL10, IL23 and IFN γ protein production in DCs which was similar in WT and ApoE KO DCs. These data are shown in supplemental figure S6I and below.

7) Figure 6 and 7 display very interesting data. However, it would be further strengthened if in figure 6K the reverse experiment would also be performed: culture ApoE2/E3 DCs with T cells from ApoE2, -3 and -4 donors. Of there is no phenotype there, it would strengthen the claim that also in humans the ApoE effect on immune cell activation is driven by changes selectively in DCs.

In response to this interesting suggestion, we investigated T cell subsets polarization in MLR experiments, incubating i) ApoE2 MDCs with ApoE2 CD4⁺, ApoE3 CD4⁺ apoE4 CD4⁺; ii) ApoE3 MDCs with ApoE2 CD4⁺, ApoE3 CD4⁺ apoE4 CD4⁺. Experiments with the same apoE isoform were performed with allogenic T cells (different donors with the same genotype). There data are presented below.

Minor comments:

1) Line 171-174: The authors use the terms conventional versus lymphoid DCs to refer to CD11b⁺ and CD8⁺ splenic DCs. However, that is an outdated nomenclature. Better use conventional (c)DC1 and cDC2, respectively.

We have revised the terms accordingly.

2) Although the approach to come to the conclusion that the phenotype can be largely explained change in DC biology is valid, it is somewhat indirect. Could the authors comment on why they have not used CD11c-Cre / ApoE flox mice?

Unfortunately, apoE floxed mice are not currently available, therefore perform experiment in this animal model will require the generation of a novel genetically modified mouse strain, which will take an amount of time far beyond the allowance for revision. This elegant approach is planned in our future work.

3) The title of Fig 2 is incorrect. There is no data in this figure that point towards a role of myeloid deficiency of ApoE in promoting rejection. Only that it is haematopoietic.

We apologize for the confusion, the word has been corrected.

4) Using the word 'Defines' in the title seems a little too strong – as if that is the only factor. I would suggest to change that to 'regulates' or 'control'.

The title has been changed in “controls” according to your suggestion.

Reply to comments from Reviewer #2

Dear Reviewer #2, in this revised version of the paper, we addressed all comments raised by you and by the other reviewers. More in details we have 1) performed antigen presentation experiments by pulsing DCs with OT-I or OT-II peptides and tested in vitro proliferation of OTI and OTII cells from the two transgenic animal models to specifically address MHC-I or MHC-II dependent antigen presentation; 2) improved the characterization of activated DCs in apoE KO mice and investigated the cytokine signature and expanded the analysis of the sterols signature by performing a more comprehensive analysis of genes and proteins related to cholesterol/oxysterol metabolism; 3) tested the distribution of intracellular vs membrane MHC-II and investigated the expression of other lipid raft-associated receptors, such as TLR4; 4) expanded the metabolic characterization of human MDCs and assessed the expression of genes related to cholesterol/oxysterol metabolism; 5) investigated functional differences in DC and T cells from carriers of the apoE2, apoE3 and apoE4 isoforms.

Regarding your specific comments, we have addressed these as outlined below:

Major:

1) The consequence of ApoE-deficiency on cellular sterol homeostasis is somewhat perplexing. The increased level of total cellular cholesterol and of several species of oxysterols (fig 5 and S7,S8) is not reflected in SREBP- and/or the LXR-regulated gene program (Fig 5c; Note: cd36 does not respond to cholesterol and the comment alluding to this in the text should be corrected). Increased accumulation of these lipids is expected to significantly decrease SREBP targets (HMGCR is not reduced) and increase LXR targets (ABCA1 and G1 are not increased). The authors invoke the argument that desmosterol is reduced, however, as shown in fig 5A and S7 this is not significant so this cannot explain the lack of change in gene expression. The authors should therefore evaluate a larger set of target genes and also consider determining the protein levels of a select number of these targets in order to draw a clearer conclusion on this. In line with this point, does pharmacological LXR activation reverse lipid accumulation, inflammatory signaling and hyperactivity of ApoE null DCs? This seems relevant since it will address the role of cholesterol accumulation and of ApoE - an LXR target - in the DC phenotype.

To address this important point, we have now investigated the expression of DHCR24, DHCR7 and CH25H. These data are now presented in Figure 5C-5D and Supplemental Figure 7B and below. Moreover we have expanded the analysis in new samples and observed also a significant reduction in HMGCoA-R expression in ApoE KO DC. The comment on CD36 has been amended accordingly.

We also investigated the impact of LXR activation on DCs lipid rafts enrichment and hyperactivity and showed that incubation with GW3965 (1 μ M) decreased lipid rafts enrichment in WT DCs to a larger but not significant different extent compared to ApoE KO DCs. Accordingly surface expression of MHC-II was reduced to a similar extent in WT and ApoE KO DCs. These data are in line with a previous work on macrophages by Ito A et al (eLife2015;4:e080009) which showed that pharmacological activation of LXR represses inflammatory gene expression in macrophages via a ABCA-1 dependent/ApoE independent mechanism. These data are shown below.

2) The analysis of lipid rafts in ApoE-KO DCs is limited. Given that in the study it is reported that ApoE KO DCs have higher MHC-II and lipid rafts (as assessed by CTXb staining) it is not surprising that when using a threshold in their analysis one would find an increased co-localization coefficient. Addressing causality may be difficult, but at the very least the authors need to evaluate whether the effect is MHC-II-specific, or that other lipid-raft associated signaling events are altered in the ApoE KO cells. For example, are surface TLRs or RTKs and their associated signaling also changed?

To address this important issue we have now a) tested MHC-II specificity by using OT-I and OT-II T cells isolated from transgenic mice and incubated them with DC pulsed with OT-I or OT-II peptides, b) investigated TLR4 expression in WT and ApoE KO DCs.

First we investigated the ability of WT and apoE KO DCs in presenting OT-II peptide or OT-I to transgenic OT-II T cells or to transgenic OT-I T cells, confirming the impact of apoE on MHC-II but not MHC-I dependent antigen presentation. This finding fits with the observation that MHC-II but not MHC-I expression is increased in ApoE KO DCs compared to WT DCs and is in agreement with a

previous study showing that MHC-II but not MHC-I has a preference for membrane GM1⁺ raft environment (Knorr R et al J Cell Science 2009; 122,1584-1594). Vice versa, in line with increased MHC-II expression, also TLR4, which is located in lipid rafts, resulted increased in DC from ApoE KO mice compared to WT. These data are presented in Figure 4D and 4E, in Figure 5F and below.

3) The attempt to translate the mouse-based findings to the human setting is commendable yet the analysis of the human DCs is rudimentary. That ApoE influences Cd4 T-cell polarization in the MLR assay (Fig 6K) is convincing. However, the evidence that this is related to the sterol-status of these cells, or to ApoE-dependent lipid transport in lacking. Are cellular sterol levels affected in the DCs in an ApoE-isoform dependent manner? Is expression of cholesterol-related genes altered? Is bona fide cholesterol efflux different in these DCs?

Thank you for your suggestions, we have investigated cellular sterol levels in apoE2, apoE3 and apoE4 DCs. This analysis showed an increased cholesterol accumulation in DCs from apoE4 carriers compared to those of apoE3 and apoE2 carriers (See Supplemental Figure 10 and below).

Furthermore, we tested the ability of ApoE3 vs ApoE4 serum to improve the phenotype in ApoE4 DC by evaluating changes in cellular sterols, lipid rafts, TNF alpha expression, HLA-DR surface expression and T effector memory cells polarization. These data are shown in figure 7 and below.

Minor:

- The article will benefit from English proof reading as well as paying attention to punctuation and use of consistent tense (also in methods and sup. Material sections). This will make readability easier. Here and there one comes across typos. For example: in Figure 1F “deplition” > depletion. Please check.

Thank you for raising this point. We have revised the manuscript accordingly.

- The figure legends are concise to the point that it’s not easy to follow some of the experiments. Some experimental information would be helpful. Also, the legends should clearly state the group/sample size as this is not always indicated.

As per your suggestion we have amended figure legends.

- **The statistics section in the Materials and Methods indicates that paired analysis was used, while the legends report unpaired. Please check and clarify.**

Sorry for the confusion, the word has been corrected with unpaired Ttest in the Material and Method section.

- **It would be informative if the authors comments on the Cd4/8 differential effect of ApoE in DCs.**

Thank you for your suggestion, we have implemented the comment on CD4/CD8 differential effect of apoE KO DCs in the text (see result section, page 9 line 4-6; discussion section, page 16 line 1-2). In detail, we attribute the different effect of apoE KO DCs on CD4 or CD8 response to apoE-mediated cholesterol mobilization which in turn shapes lipid raft composition, where predominantly MHCII but not MHCI proteins are located. Please see pages 7 to 10.

- **In the graft experiments, a cut-off of 50% skin-surface rejection is considered as a full rejection. Please clarify the rational for using this (arbitrary) cut-off value.**

Allograft rejection was determined macroscopically and to harmonize data presentation we perform a yes-or-no analysis scoring full rejection when necrotic donor skin was more than the 50%.

- **Figure 5: scale bars in IF images are missing.**

Scale bars are now included.

- **The in vivo EaGFP procedure is missing in the methods section. Please add.**

Thank you for raising this point. We have added EaGFP procedure in material and methods section.

- **The paragraph discussing LXR in the Discussion section is unclear and confusing (lines 283-98) and should be clarified. Furthermore, the argument invoking the role of ABCA1 as an anti-inflammatory determinant of LXR activation should take into account results from the Tall group that find that LXR activation is anti-inflammatory also in Abca1/G1-null cells.**

We agree with your comment as many works highlighted an LXR-dependent anti-inflammatory effect beyond cholesterol modulation and modified the discussion as follows: "The anti-inflammatory effects of LXR were recently proposed, at least in macrophages, to be the consequence of ABCA-1-dependent changes in membrane lipid organization but not of LXR dependent ABCG1 or apoE modulation (Ito A, eLife 2015). However, an LXR-dependent repression of inflammatory gene expression in activated macrophages has been proposed beyond cholesterol

modulation, through the inhibition of multiple NF- κ B target genes (COX-2, I κ B α , IL-1 β , MCP-1, IL-6 and IL1-receptor antagonist) after LPS, TNF- α , or IL-1 β stimulation (Joseph SB, Nature Medicine 2003).”

- **Figure S8A,B is not clear. What is being compared and between which cells?**

The correlation matrix was used to investigate the dependence between phospholipids, fatty acids, sterols and oxysterols within CD11c⁺ isolated from the spleen of WT or apoE KO mice. This information has now been added for clarity to the legend of Supplemental Figure 8.

- **L.130: The statement made in this line seems wrong since hypercholesterolemia is largely corrected in the KO mice receiving WT marrow. Please correct or clarify.**

We apologize for the confusion and have corrected the sentence.

- **The change in color coding of the legend in fig 7 is confusing.**

Color coding in Figure 7 has been changed to make readability easier.

Reply to comments from Reviewer #3

Dear Reviewer, in this revised version of the paper, we addressed all comments raised by you and by the other reviewers. More in details we have 1) performed antigen presentation experiments by pulsing DCs with OT-I or OT-II peptides and tested in vitro proliferation of OTI and OTII cells from the two transgenic animal models to specifically address MHC-I or MHC-II dependent antigen presentation; 2) improved the characterization of activated DCs in apoE KO mice and investigated the cytokine signature and expanded the analysis of the sterols signature by performing a more comprehensive analysis of genes and proteins related to cholesterol/oxysterol metabolism; 3) tested the distribution of intracellular vs membrane MHC-II and investigated the expression of other lipid raft-associated receptors, such as TLR4; 4) expanded the metabolic characterization of human MDCs and assessed the expression of genes related to cholesterol/oxysterol metabolism; 5) investigated functional differences in DC and T cells from carriers of the apoE2, apoE3 and apoE4 isoforms.

Regarding your specific comments, we offer the following replies:

1. The conclusion that allograft rejection is independent of a systemic inflammatory response (line 126) is based on cholesterol levels (Fig 2C), which do not appear to correlate with allograft rejection (Fig 2B). This is indirect and evidence of similar levels of inflammatory mediators would be required to demonstrate independence of systemic inflammatory responses in allograft rejection, especially since irradiation and the allo reaction itself are also likely contributing to systemic inflammatory responses in these models.

To further support the conclusion that allograft rejection is independent of systemic inflammatory response we have analyzed markers of CD4⁺ T cell differentiation in the blood of grafted BMT. As shown, levels of T effector memory cells (TEM) and activated (CD25⁺ and CD44^{hi}) CD4⁺ T cells were increased in the circulation of apoE KO mice independently of being transplanted with KO or WT BM (blu and orange bars), compared to WT mice transplanted with WT or KO BM (white and light blu bars). WT mice transplanted with KO BM (KO BM to WT mice) presented a faster rejection, similar to that of KO BM to KO mice, but showed a profile of circulating CD4⁺ Tcell similar to that of WT BM to WT mice; thus suggesting that allograft rejection is not driven by systemic immunoinflammation. These data are now presented in Supplemental Figure 4B-4D and below, and are discussed in the result section, page 6 last paragraph.

2. Fig 2E shows that ApoE deficiency within the BM compartment rather than the T cells themselves are important. The Fig 2 title should state “BM deficiency of ApoE” since a direct role for myeloid cells has not been shown. Likewise “an immunomodulatory role for myeloid but not hepatic apoE” (abstract lines 28-29) is not well supported by the data. Cells other than myeloid cells are derived from transferred BM and myeloid cells in the liver may also be contributing.

We agree with your comments; the title of figure 2 and the abstract have been revised accordingly.

3. ApoE ko mice more readily reject allografts (Fig 1F) and conversely, ApoE ko grafts are more readily rejected (Fig 3J), suggesting ApoE deficiency in both donor and host are important in allograft rejection. However, these data do not demonstrate a direct role for DC in graft survival and hence the conclusions that “lack of apoE expression by graft-derived DC was responsible” (lines 162,163) is not supported by the data. Demonstration of a role for DC in this model would require a strategy (eg depletion Abs, DTR mouse models) to remove DC.

Thank you for bringing this point to our attention, we have reformulated the sentence as follows “As show in in Figure 3J, apoE deficiency in the skin graft significantly reduced graft survival, indicating that lack of apoE expression by graft-derived antigen presenting cells might be responsible for enhanced allograft rejection.” (page 8 line XXX)

4. Fig 3 C-G: What is CD4 mitotic index?? How was this calculated??

Mitotic index is defined as the ratio between the number of Tcells undergoing mitosis to the total number of plated T cells. The index is calculated at each cell division from the sum of mitotic events adjusted for the number of cell precursors. This information has now been added to material and method section.

5. ApoE ko mice have increased numbers of DC and increased MHC II expression. However they also have significantly lower levels of activation marker CD80 (Fig S6) which is at odds with the statement that ApoE deficiency is not associated with costimulatory function (line 178). In order to make this conclusion expression of costimulatory molecules and DC cytokine secretion should be examined in the presence or absence of TLR ligands or other well-defined DC stimuli.

We agree with this point and performed a new set of analyses on CD80 expression on cDC2 cells in the spleen of WT and apoE KO mice. As shown below in this experimental setting, the expression was similar in apoE KO and WT DCs. These data are known presented below and in Supplemental Figure 6D and are discussed in the text at page 9, second sentence.

D

Next we investigated CD86 and CD80 expression in BMDCs WT and ApoE KO during maturation (GM-CSF for 6 days) followed by incubation with LPS (10ng/mL, 18h). In a similar experimental setting we investigated INF γ , IL-12, IL-23 and IL-10 expression upon LPS stimulation (10 ng/mL, 4h). These data are presented in Supplemental Figure 6E-6I and below.

E

F

G

H

I

6. Details of the ag uptake and processing assay are missing from the methods and the dot plots in Fig 4 are not convincing. The percentage of GFP+ α :IAb+ events for both WT and ApoE KO are <2% in the representative dot plots in Fig 4G yet over 10% in all of the data points in Fig 4F. Relevant control plots should be shown. The inclusion of other assays such as DQ-OVA fluorescence and or and processing and presentation of Ag to Ag-specific T cells would provide more convincing evidence for a direct role of ApoE in antigen processing and presentation.

To address this issue we show below the entire panel with the relevant control plots from the new experiments performed in WT and ApoE KO cells while a representative panel is presented in Figure 4G.

Moreover, we have investigated the ability of WT and APOE KO DC in presenting OT-II peptide or OT-I to transgenic OT-II T cells or to transgenic OT-I T cells, confirming the impact of apoE on MHC-II but not MHC-I dependent antigen presentation. These data are now presented in Figure 3 panels F to H and below.

7. Fig 5C what is the fold of expression relative to?

We apologize for the omission, the axis refers to mRNA expression relative to CD11c⁺ isolated from WT spleen and has been specified in Figure 5C.

8. The conclusions that increased CD4⁺ TEM polarization in humans carrying the apoE4 isoform is due to enhanced antigen presentation activity by DC (line 245) is not well supported by the experimental design shown in Fig 6J. In this experiment, autologous MDC and naïve CD4⁺ T cells are co-cultured in the absence of exogenous antigen. Under these conditions T cells would not be expected to proliferate at all and both T cells and DC would be expressing the same ApoE4 isoform. It therefore isn't clear whether the T cell phenotype changed during the culture and therefore whether autologous DC in the steady state had any influence on this. To test whether antigen uptake and processing is altered by the different isoforms expressed by MDC DQ-OVA or similar functional assays would need to be performed. For CD4⁺ T cell proliferation and differentiation of effector function, MDC from donors expressing different ApoE isoforms would need to be co-cultured with allogeneic CD4 T cells, preferably from a donor expressing the parent ApoE3 isoform. Performing these experiments in the presence and absence of the apoE3-HDL would further increase the quality of the data.

We apologize for the confusion, experiments presented in Figure 6J and 6K were performed by pulsing DC with LPS (1 µg/mL) for 2 days, followed by coculture with autologous T CD4⁺ T cells for 5 days. This is now clarified in Figure 6J and in the text. Furthermore, as per your suggestion we performed new MLR experiments by pulsing DC from ApoE2, ApoE3 or ApoE4 donors for 2 days with LPS (1 µg/mL) and cocultured with the same allogeneic T cells from apoE donor for 3 days. Incubation with apoE4 DC results in increased TEM polarization was observed in T cells from an apoE3 donor. These data are now shown in Supplemental Figure 10C and below.

We next explored the impact of different ApoE-HDL isoforms. To this aim MLR experiments were performed by pulsing DC from ApoE4 donors for 2 days with LPS (1 µg/mL) and then culturing with the allogeneic T cells from an apoE3 donor for 3 days in the presence of serum from ApoE4 or

ApoE3 donors different from those of DC or T cells (allogenic). Interestingly these data suggest that ApoE3 HDL- may partially revert the phenotype. These data are shown in Figure 7I and below.

9. The data in Fig 7 using human MDC shows % of DC expressing HLA-DR and CD80. All DC should express HLA-DR. Representative stains, gating strategies and isotypes or FMO controls should be shown in the supplementary data. The data in the mouse models shows changes in the intensity of MHC Class II and CD80 expression rather than percentage of expressing cells. A similar analysis of the human data could be informative.

To address this issue we have included the intensity of HLA-DR and CD80 in human MDC in Figure 7 and a representative panel of histograms? Dot-plots, gating strategies and controls in supplemental panel 11 and below.

10. The published literature describing ApoE and the phenotype of the ApoE ko mice in discussion (lines 259-269) should be moved to the introduction.

We have changed the discussion and the introduction as per your suggestion.

REVIEWERS' COMMENTS:

Reviewer #1 (Remarks to the Author):

The authors have performed a sizeable number of additional experiments that have addressed all my concerns satisfactorily. As a result, the manuscript is significantly improved and I have no further comments.

Reviewer #2 (Remarks to the Author):

I want to thank the authors for addressing the comments that were brought up.

One small question that remains relates to Fig 7G. The authors were asked about expression of cholesterol-related genes in DCs from the different APOE isotopes. This is not directly addressed, and is relevant particularly as the sterol analysis suggests that the isoform-specific alteration in sterol species extends beyond cholesterol (desmosterol and other oxysterols are also increased in E4>E3>E2 manner; are these significant?). In line with, the effect of the E3 and E4 serum on gene expression seems inconsistent and variable. This may be due to the fact that MDCs from only two donors were tested. The authors should comment on this.

- Legend of Supp fig. 10: Spleen-derived DCs? Please check.

Reviewer #3 (Remarks to the Author):

The revised manuscript contains a significant amount of new data that adequately addresses the issues raised and considerably strengthens the main messages. The following are minor points to be addressed:

1. Line 30 "produced by BM cells"- change to "BM -derived" or "cells of haematopoietic origin"
2. IFN γ is mis-spelt as "INF" throughout
3. The addition of data showing enhanced proliferation of OTII but not OTI cells using ApoE ko DC (Fig 3) strengthens the findings, but it would have been informative to have also included OVA protein as antigen.
4. The colours of the different histograms in Figs 3 F,G, 4E,H,5F need to be changed to more clearly distinguish them.
5. Lin 158 "pulsed with OTI or OTII peptide" should be "pulsed with peptides specific for OTI or OTII T cells".
6. Line 162 "Moreover, T cells...." Should specify which T cells this refers to.
7. Lines 190 and 192 "T cell" should be "T cells". There are numerous other grammatical errors throughout which require attention.
8. New data in Fig 5F shows increased TLR4 expression in KO DCs. This increase could be expected to increase costimulatory function and cytokine secretion in response to LPS. However, this does not appear to be the case, at least for cDC2 (Fig S6). The LPS responsiveness of the BM DC (co-stimulatory molecule expression, cytokine secretion) needs to be examined and the TLR4 expression confirmed in cDC2. At the very least the discrepancy between TLR4 expression levels and LPS responsiveness by the WT and KO DC needs to be discussed.